# Snow Cover Reconstruction in the Brunswick Peninsula, Patagonia, Derived from a Combination of the Spectral Fusion, Mixture Analysis, and Temporal Interpolation of MODIS Data

**Francisco Aguirre** [1,2,3,*] ⬥, **Deniz Bozkurt** [4,5] ⬥, **Tobias Sauter** [6] ⬥, **Jorge Carrasco** [1] ⬥, **Christoph Schneider** [6] ⬥, **Ricardo Jaña** [7] ⬥ **and Gino Casassa** [1]

1   Centro de Investigación Gaia Antártica (CIGA), Universidad de Magallanes, Punta Arenas 6210427, Chile; jorge.carrasco@umag.cl (J.C.); gino.casassa@umag.cl (G.C.)
2   Ecology and Biodiversity Institute (IEB), Puerto Williams 6350080, Chile
3   The School for Field Studies, Center for Climate Studies, Puerto Natales 6161009, Chile
4   Center for Climate and Resilience Research (CR)2, Santiago 8320000, Chile; deniz.bozkurt@uv.cl
5   Departamento de Meteorología, Universidad de Valparaíso, Valparaíso 2340000, Chile
6   Geography Department, Humboldt-Universität zu Berlin, 12489 Berlin, Germany; tobias.sauter@geo.hu-berlin.de (T.S.); christoph.schneider@geo.hu-berlin.de (C.S.)
7   Scientific Department, Chilean Antarctic Institute, Punta Arenas 6200965, Chile; rjana@inach.cl
*   Correspondence: francisco.aguirre@umag.cl

**Abstract:** Several methods based on satellite data products are available to estimate snow cover properties, each one with its pros and cons. This work proposes and implements a novel methodology that integrates three main processes applied to MODIS satellite data for snow cover property reconstruction: (1) the increase in the spatial resolution of MODIS (MOD09) data to 250 m using a spectral fusion technique; (2) a new proposal of snow-cloud discrimination; (3) the daily spatio-temporal reconstruction of snow extent and its albedo signature using the endmembers extraction and spectral mixture analyses. The snow cover reconstruction method was applied to the Brunswick Peninsula, Chilean Patagonia, a low-elevation (<1500 m a.s.l.) mid-latitude area. The results show a 98% agreement between MODIS snow detection and ground-based snow measurements at the automatic weather station, Tres Morros (53.3174°S, 71.2790°W), with fractional snow cover values between 20% and 50%, showing a close relationship between snow and vegetation type. The number of snow days compiled from the MODIS data indicates a good performance (Pearson's correlation of 0.9) compared with the number of skiing days at the Cerro Mirador ski center, Punta Arenas. Although the number of seasonal snow days showed a significant increasing trend of 0.54 days/year in the Brunswick Peninsula during the 2000–2020 period, a significant decrease of −4.64 days/year was detected in 2010–2020.

**Keywords:** snow cover; remote sensing; spectral downscaling; Patagonia

## 1. Introduction

Snow cover is a critical component of the cryosphere interlinked with climate at local, regional, and global scales [1]. Through radiative and thermal properties, various Earth system components modulate the transfer of energy and mass at the Earth surface–atmosphere interface [2]. Snow-covered areas at lower elevations (<ca. 1500 m a.s.l.) in mid-latitudes are sensitive to climate change [3,4] since the surface temperatures are close to the melting point. Various feedback processes, such as the albedo effect, thermal insulation properties, and meltwater input, make this sensitivity even stronger [5]. Changes in the seasonal behavior of snow cover, such as melting patterns, soil moisture, and water availability, can strongly affect both plant communities as well as the hydrological cycle of watersheds and their ecosystems [6–8].

Snow appears bright to the human eye because of its high and relatively flat reflectance in the visible region of the electromagnetic wavelength (400–700 nm), while it has higher absorption in parts of the near- and mid-infrared ranges [9,10]. However, the pattern of the spectral response of snow depends on its density, grain size, liquid water content, and the presence of impurities, such as dust, soot, and algae [9]. Several optical remote sensing data (such as AVHRR, MODIS, VIIRS, and Landsat) can represent the snow cover extent in different temporal, spectral, and spatial resolutions [9]. For instance, the Landsat suite of sensors, with a spatial resolution of 30 m in visible, near-, and short-infrared bands, is ideal for snow applications in mountains. The spectral bandwidth of Landsat data allows for accurate retrievals of fractional snow-covered areas ("fSCs", as a subpixel fraction) [11–13]. However, the repeat-pass interval of 16 or 18 days is not adequate for short-term snow-mapping requirements [14], especially in cloudy regions. MODIS (moderate-resolution imaging spectroradiometers), instead, with a spatial resolution of 250 m in 2 bands and 500 m in 5 visible and near-infrared bands, has monitored snow cover at a daily temporal resolution since 2000, employing an automated snow cover algorithm developed at the Goddard Space Flight Center (GSFC (http://modis-snow-ice.gsfc.nasa.gov), accessed on 1 December 2021) [14–16]. Standard MODIS snow map products include fSC, snow albedo, and cloud-gap filling. Improvements to the algorithm are incorporated each time the entire dataset is reprocessed [14], which has currently been updated to collection version 6 (https://nsidc.org/data/MOD10A1, accessed on 8 August 2023). The basic index (Equation (1)) for generating an operational snow cover product from the MODIS data is the normalized difference snow index (NDSI), as described by Hall and Riggs (2011) [17], which is the most common method used for snow cover-extent identification [18].

$$NDSI = \frac{\rho B_4 - \rho B_6}{\rho B_4 + \rho B_6} \tag{1}$$

where $\rho B_4$ is the green band reflectance (bandwidth 545–565 nm) and $\rho B_6$ is a mid-infrared reflectance (bandwidth: 1628–1652 nm). A pixel with an NDSI > 0.4 is assumed to have snow cover (binary result: snow or no snow), while a pixel with an NDSI $\leq$ 0.4 is snow-free [19].

This global snow cover algorithm has been subjected to several refinements over the years and different thresholds have been incorporated into its general formula [14]. Nevertheless, it is widely known that the NDSI has problems in differentiating snow from clouds [15,19–21]. Consistent results have not yet been achieved in the actual collection (version 6) and the problems of the previous version remain [16].

Spatial resolution is a factor that must also be considered to assess the uncertainties of remote sensing products in estimating snow cover. Hydrological studies and snow model validations in a mountainous terrain need the greatest possible spatial detail, especially when snow interaction with vegetation is relevant [22,23]. This allows a better representation of snow season behavior, such as the snow line [21], and also a more accurate discrimination of snow albedo [13,24]. Efforts to improve the spatial resolution of snow products are thus highly desirable. For snow map studies in a complex terrain, The Global Climate-Observing System [25] recommends a spatial resolution lower than 100 m, while in a flat terrain, 1 km is considered adequate. In the case of a complex terrain, snow distribution can be assessed using different techniques and sensors, for example, a combination of 30 m-resolution Landsat and a 16-day repeat pass with MODIS at a 250 m resolution and a daily temporal resolution.

The development of accessible and comprehensive methods to reconstruct snow properties using satellite remote sensing data is an emerging topic, particularly regarding its interaction with vegetation. For instance, Sirguey et al. (2009) [21] developed a comprehensive method to produce regional maps of New Zealand for the snow cover extent at a 250 m spatial resolution, using the MODIS Level-1B data product with corrected reflectivity. They showed that subpixel snow fractions could be retrieved with a mean absolute error (MAE) of 6.8% at 250 m. Painter et al. (2009) [12] improved products from MODIS (MOD09GA)

by applying a combination of atmospheric corrections and a physical algorithm called MODSCAG (MODIS snow-covered area and grain size), which yielded an average RMS error of 5% for snow cover at a spatial resolution of 500 m. MODSCAG has the advantage of detecting different kinds of snow at the subpixel level and, thus, the albedo. In comparison, the MOD10 snow albedo product (version 5) has an MAE about three-times higher, and no improvement has been made to the actual collection (version 61) [16]. The advantages of discriminating snow cover extent, snow grain size, and albedo are very important, considering that they are critical information for studies on energy and mass balance [12,26]. However, many recent studies are still using classical MOD10 products to both reconstruct snow cover variability [27,28] and validate snow models [29,30].

Southern Patagonia is a key location for climate change studies, particularly spatio-temporal snow cover variability, due to its unique and continuous extent in the Southern Hemisphere beyond 45°S [31], reaching 56°S at Cape Horn [32]. Southernmost South America is considered a real "hotspot" of paleoclimatic records, which have recorded, on a scale of hundreds to thousands of years, abrupt changes in circulation regimes and atmospheric temperatures associated with global climate variability, including ice cores, geomorphological features, peat bog systems, marine and lake core sediments, and tree rings [33–38].

The climate and topography interactions in southernmost South America are largely dominated by westerly winds occurring between the high-pressure center of the Southeast Pacific Ocean (SPH) and the circumpolar low-pressure belt surrounding Antarctica [39]. The southern annular mode (SAM) is the main mode of large-scale variability that controls the strength of the westerlies [40]. These westerlies are disturbed by both the southern and Fuegian Andes (or Darwin Cordillera), creating one of the most extreme precipitation gradients on Earth [41–43]. A recent study based on a 17-year-long meteorological record of a west–east longitudinal transect recorded an extreme precipitation gradient from 6100 mm/year at Paso Galería station, located in Gran Campo Nevado (52°45′S; 73°01′W), to 590 mm/year in Punta Arenas (53°08′S; 70°53′W). In other words, in a west–east transect of only 150 km, the annual precipitation decreased by more than one order of magnitude [44].

Snow cover studies and field stations in southern Patagonia are limited. A significant decrease of 19% in the snow cover extent over an area of 1637 km$^2$ covering a large part of the Brunswick Peninsula (~53°S), Patagonia, was detected in the last 45 years (1972–2016 [3]), attributed to a significant increase in winter temperatures, estimated as 0.71 °C at the weather station of Punta Arenas (Figure 1).

Due to the sparse observational network and uncertainties in the remote sensing snow products, current efforts and studies do not allow us to perform a quantitative and comprehensive assessment of snow cover variability in this region. Therefore, the main objective of this work is to seek a comprehensive and robust method for the spatio-temporal reconstruction of snow cover in southern Patagonia that may be replicated in other areas for comparative studies, establishing baselines for future snow and hydrologic modeling in this remote region. We address the following questions: (1) what is the best feasible method for improving the spatial resolution of the MODIS product? (2) What snow-detection methods based on remote sensing are the most adequate for southernmost Patagonia? (3) How can a daily spatial–temporal reconstruction of the snow cover properties be implemented considering a large number of cloudy days in Patagonia? (4) How has the spatio-temporal snow cover variability evolved since 2000?

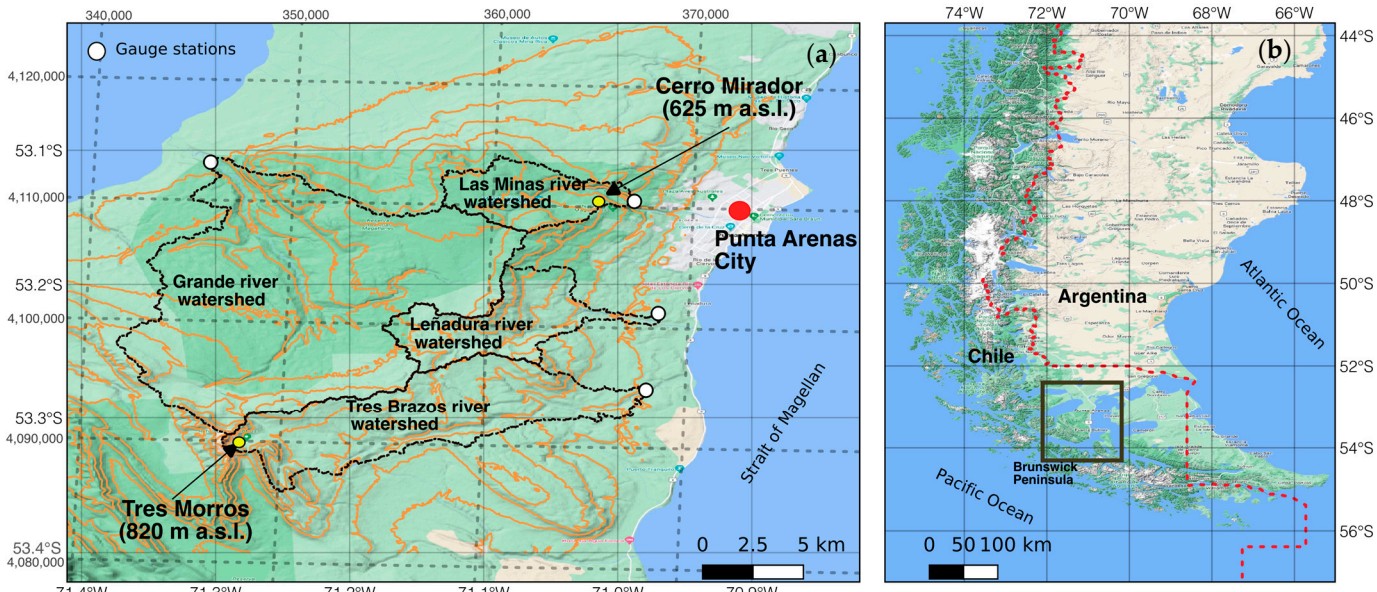

**Figure 1.** (**a**) Study area within the Brunswick Peninsula, showing the 4 selected basins in latitude/longitude and UTM 19S WGS84 coordinates. White circles are runoff stations and yellow circles are referential sites for ski activities (Cerro Mirador and Tres Morros). Contour line interval (in orange) is 100 m. (**b**) Reference location of the study area in southern South America. Figure designed using open-source software QGIS v_3.16 (QGIS Geographic Information System. QGIS Team (2017). QGIS Geographic Information System. Open Source Geospatial Foundation Project. Available Online at: https://qgis.org, accessed on 29 October 2020).

## 2. Study Area

The Brunswick Peninsula is located in southwest Patagonia (53°S, 71°W). Vegetation includes deciduous forest (Nothofagus pumilio and Nothofagus antarctica), peatland (mosses of the genus Sphagnum), grassland, and shrubs (i.e., Lepidophyllum cupressiforme, Empetrum rubrum, and Gaultheria mucronata). The climate has a strong oceanic influence, being dominated by westerly year-round circulation. The city of Punta Arenas, located in the north-eastern sector of the Peninsula, concentrates 80% of the population of the Magallanes province, one of the areas with the lowest population densities in Chile (1.1 inhabitants/km$^2$) (http://www.subdere.gov.cl/división-administrativa-de-chile/gobierno-regional-de-magallanes-y-antártica-chilena, accessed on 20 December 2020). The water balance in the Brunswick Peninsula is critically important regarding the water supply to the population. In addition, the presence of seasonal snow has allowed the development of the ski resort "Club Andino", in 1938, located in Cerro Mirador, at an elevation of 350–625 m, 8 km from the Punta Arenas city center. Snow cover has shown a relevant reduction over the last several decades, mainly due to atmospheric warming [3], significantly reducing the operation of the ski center, which only managed to open for a period of less than 1 month in 2016–2019 (Table 1), whereas until the 1990s, the skiing season typically lasted for over 2 months (Rodrigo Adaros, personal communication). The lack of snow at Club Andino has caused the regional government to explore an alternative location at Tres Morros hill, 31 km southwest of Punta Arenas city, at an elevation of 550–820 m, as a possible alternative site for the development of a future skiing area.

**Table 1.** Club Andino's operational days in 2010–2020.

| Year | 2010 | 2011 | 2012 | 2013 | 2014 | 2015 | 2016 | 2017 | 2018 | 2019 | 2020 |
|---|---|---|---|---|---|---|---|---|---|---|---|
| Season length | 78 | 85 | 65 | 76 | 57 | 95 | 10 | 0 | 30 | 30 | Closed due to pandemic |

The study area in the Brunswick Peninsula is defined by 4 basins, each of which has a river runoff station at the outlet, operated by Dirección General de Aguas (DGA, General Water Directorate of Chile). The basin boundary polygons were obtained from the CAMELS-CL dataset [45] (Figure 1).

## 3. Data and Methods

In this section, we describe the different data and a novel methodology used to reconstruct the spatio-temporal variability of snow in southernmost Patagonia. The data included MODIS sensor Terra satellite products and the meteorological records of automatic weather stations, as well as an atmospheric re-analysis and regional climate model outputs. All calculations were performed using the open-source programming language R v._3.6.3 (https://www.r-project.org, accessed on 8 August 2023). Figure 2 shows a flow diagram of the different algorithms and processes evaluated and implemented in the present work. The novel characteristic of the methodology developed here is the semi-automatic integration of three main methods: spatial downscaling spectral fusion, spectral mixture analysis, and spatial–temporal snow cover reconstruction. One of the main improvements was the snow-cloud discrimination (Section 3.2.3), where we proposed a snow mask based on a combination of three thresholds for snow detection. Each of these processes must be conducted sequentially using the different scripts provided in the Supplementary Materials and codes.

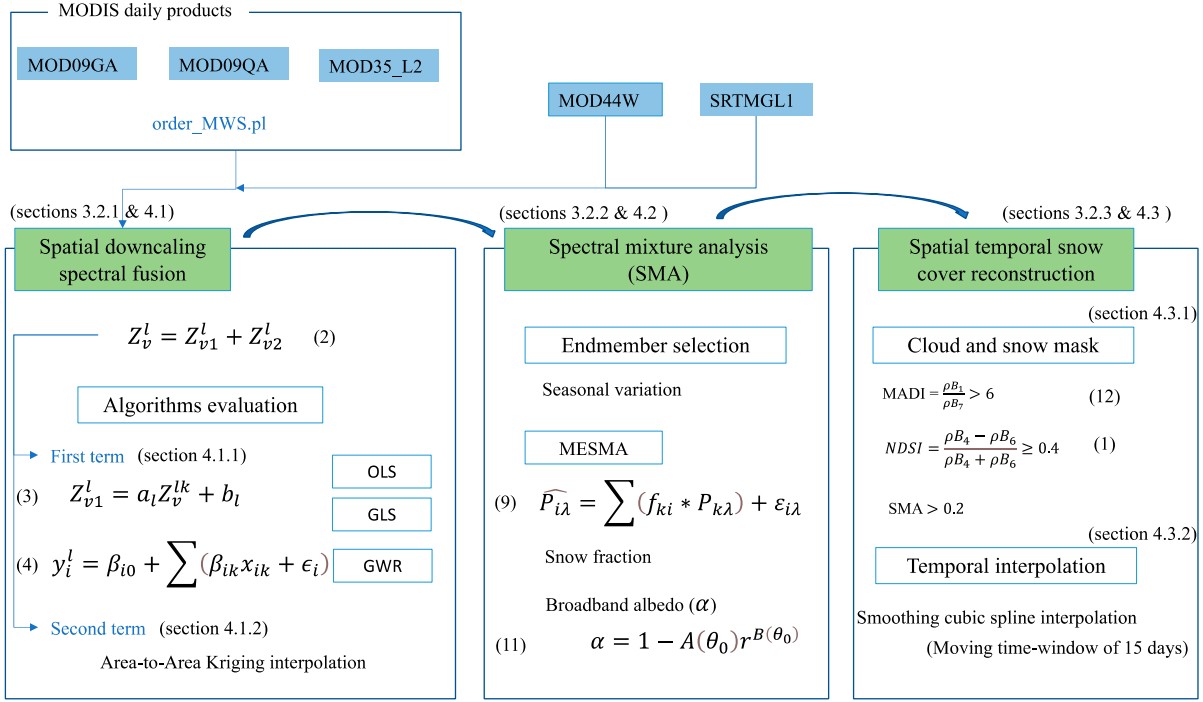

**Figure 2.** General diagram of the snow cover reconstruction method.

### 3.1. Data

#### 3.1.1. Satellite Sensor Data

The basic information for snow cover reconstruction was obtained from moderate-resolution imaging spectroradiometer (MODIS) sensor data at a resolution of 500 m (3 visible (RGB), 2 near-infrared, and 2 mid-infrared bands) and 250 m (1 visible (red) and 1 near-infrared band) of MOD09GA and MOD09QA land products version 6 (including atmospherically corrected reflectance), at daily temporal resolutions (Figure 3). In addition, MODIS products MOD35_L2 (daily cloud mask at a 1 km resolution) and MOD44W version 6 (water/land mask at a resolution of 250 m), available from the Land Processes Distributed Active Archive Centre (http://lpdaac.usgs.gov/, accessed on 8 August 2023) were used.

MOD44W data were downloaded from https://lpdaac.usgs.gov/products/mod44wv006/, accessed on 8 August 2023 and exported to a Geotiff format with the software HEG:HDF-EOS v.2.15 (available in https://wiki.earthdata.nasa.gov/display/DAS/HEG%3A++HDF-EOS+to+GeoTIFF+Conversion+Tool, accessed on 8 August 2023), adjusting the area limits for each watershed and considering the nearest-neighbor resampling interpolation. Following the previous analysis related to the snow-season duration in the Brunswick Peninsula, which showed snow cover from May to October [3], we downloaded images between April and October to reconstruct the cold-season snow variability. The topographic elevation used in this study was derived from the NASA Shuttle Radar Topographic Mission Global [46] 1 arc second (ca. 30 m) (SRTMGL1) version.003 (https://search.earthdata.nasa.gov/search?q=SRTMGL1, accessed on 8 August 2023) digital elevation model (DEM) dataset.

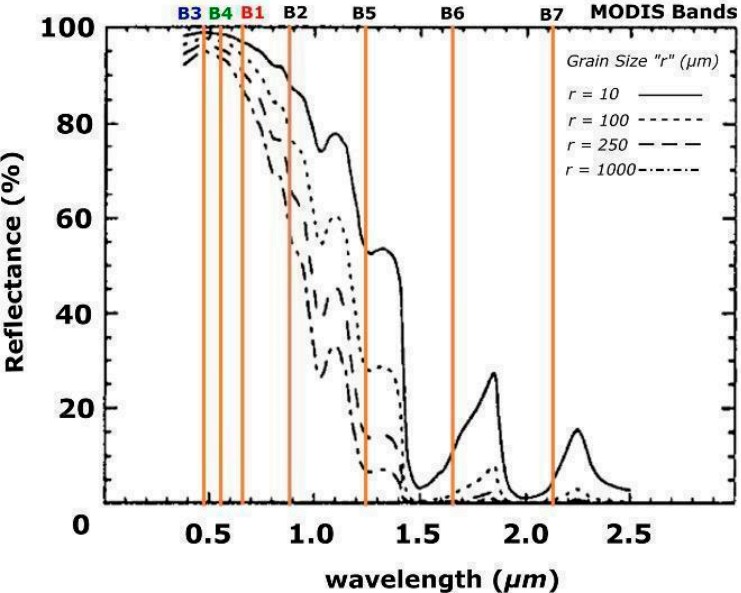

**Figure 3.** Snow spectral signatures of different snow grain sizes. MODIS bands B1 to B7 are shown as green vertical lines. Image modified from Painter et al. (1998) [47].

### 3.1.2. Weather Stations

In southernmost Patagonia, the scarcity of locally observed meteorological data in both spatial and temporal representations is a major difficulty for climate change studies. Here, we used the data from the Tres Morros automatic weather station (AWS), Brunswick Peninsula, for 27 July 2018, specifically for snow cover studies, including winter sports evaluations. This station, called AWS Tres Morros, was located at the top of the Tres Brazos river watershed (53.3174°S, 71.2790°W) at an elevation of 554 m a.s.l. The variables measured at AWS Tres Morros were snow height (m), atmospheric pressure (mb), air temperature (°C), air relative humidity (%), wind direction (°), velocity (m/s), and short- and long-wave radiation values (Wm$^{-2}$), both incoming and outgoing.

### 3.2. Methods

### 3.2.1. Downscaling and Spectral Fusion

Multimodal remote-sensing image and data fusion is a relevant research field that has increasingly developed in the recent years [48]. The goal is to obtain the most complete and accurate information with adequate spectral and spatial resolutions of the study area, given that remote sensing data normally cannot yield both high spectral and spatial resolutions at the same time. Several methods have been developed to combine the advantages of spectral and spatial resolutions in one image [49] and, in particular, regarding MODIS products, including principal component analysis (PCA) [50], wavelet transformation [51,52],

high-pass filter (HPF) [53], and kriging with external drift [54]. Of particular relevance is the work of Wang et al. (2015) [55] on downscaling spectral fusion, which improved MOD09GA at 500 m to produce higher-quality downscaled images with a 250 m resolution using a combination of linear regression and area-to-point kriging interpolation (ATPRK) applied to the residuals of the regression (Equation (2)). The method proposed by Wang et al. (2015) [55] was adjusted and implemented in this work since it showed that high-quality sharpened images could be obtained, as compared to the other commonly used methods mentioned previously. It is important to consider that these implementations and comparisons by Wang et al. (2015) [55] were performed on a single image, something common among comparisons conducted on spectral fusions, without clouds or snow.

The downscaling spectral fusion equation of Wang et al. (2015) [55] is composed of a sum of 2 terms, as given by Equation (2):

$$Z_v^l = Z_{v1}^l + Z_{v2}^l \tag{2}$$

where the first term $Z_{v1}^l$ is the result of the prediction of regression (presented in Equation (3)) and $Z_{v2}^l$ is the residual value between the regression predictions and real values for each pixel, using MODIS coarse bands at a 500 m resolution (l = (bands 3 and 7)).

The prediction of regression is as follows:

$$Z_{v1}^l = a_l Z_v^{lk} + b_l \tag{3}$$

is a linear regression model where $a_l$ and $b_l$ are the two coefficients for the l-th band and $Z_v^{lk}$ is the covariate band (bands 1 and/or 2).

Both terms were downscaled to 250 m. For Equation (3), the assumption that this relationship was universal at different spatial resolutions was made; therefore, the relationship observed for coarse spatial resolutions could be applied to the highest spatial resolution [56]. Therefore, we could use $a_l$ and $b_l$ to predict $Z_{v1}^l$ at a 250 m resolution. For the second term of Equation (2), we used a geostatistical approach to interpolate the residuals from a resolution of 500 to 250 m.

Geostatistics-based spatial scaling in the area-to-point regression (ATPRK) and area-to-area regression kriging (ATARK) has been widely used in remote sensing applications, including downscaling processes [55–57]. The most important difference between area-to-point regression and area-to-area kriging was the covariance calculation between samples where area-to-area kriging had the advantage of yielding areal values directly [58]. A recent study by Jin et al. (2018) [59] proposed to modify the linear regression with a geographically weighted regression (GWR) as a more suitable model that considered the spatial heterogeneity and non-stationarity (Equation (4)), changing the first term in the interpolation approach as follows ($Z_{v1}^l \sim y_i^l$):

$$y_i^l = \beta_{i0} + \sum (\beta_{ik} x_{ik} + \epsilon_i) \tag{4}$$

In this geographically weighted linear regression of Equation (4), each pixel (*i*) has specific regression coefficients ($\beta_{i0}$ and $\beta_{ik}$) and a covariate band ($x_{ik}$ and errors ($\epsilon_i$)).

In this work, we compared both downscaling methods, GWRs (with and without kriging residual interpolations), and ATARK, exploring the ordinary least squares (OLS) method and also incorporating a generalized least squares (GLS) method to deal with the problem of the violation of homogeneity (also called heteroscedasticy) in cases where it was detected [60]. To preserve the spatial correlation (variogram or semivariogram), we also fitted a spatial correlation model to each relationship (variogram model). The Akaike criterion (AIC) was used for model evaluations along with the coefficient of the determination ($R^2$) and root mean square error (RMSE). In contrast to the work of Wang et al. (2015) [55], here we compared the different approaches for an image with snow and another without snow, since this study focused on improving the representation of snow cover and its spectral interactions with other land-cover types.

To implement these downscaling schemes, we used the following R packages: raster [61] to manipulate all the raster variables and transformations; nlme [62] to implement and analyze linear and nonlinear mixed models; gstat [63] to implement spatial interpolations and analyses; atakrig [57] to implement ATARK; and GWmodel [64] to create the GWR model. All values out of the valid range [−100, 16,000] were filtered and a water mask at a 250 m spatial resolution was used (MOD44W).

We use different indexes for the quality assessment and selection of the downscaling methodology. To check the quality of the spectral properties for each downscaled band, we used the universal quality index (UQI), a simple and intuitive measure of spectral fidelity between two bands [49,65], with each image having the same dimension (Equation (5)):

$$UQI = 4 \frac{\sigma_{fg} \overline{f} \overline{g}}{\left( \overline{f}^2 + \overline{g}^2 \right) \left( \sigma_f^2 + \sigma_g^2 \right)} \tag{5}$$

In the UQI equation (Equation (5)), $\overline{f}(\overline{g})$ and $\sigma_f^2$ ($\sigma_g^2$) are the mean and variance of band $f(g)$ and $\sigma_{fg}$ is the covariance between the two bands f and g. This index must be implemented in a sliding window and the average index is then calculated over all these windows. For its application, we assumed that the downscaled image at a high resolution was incorporated and added to the original low-resolution image [56]. For this purpose, we applied a low-pass filter and compared the downscaled image to the original low-resolution image. The advantage of the UQI index is that it allows us to assess the quality of the downscaling by spectral fusion for each band separately.

As a complement to the UQI, we added the 'Quality Index without Reference' (QNR, Equation (6)), an index that did not require a reference image [66]. QNR is composed of UQI, the spectral distortion index (Dγ, Equation (7)), and the spatial distortion index (Ds, Equation (8)), since it evaluates the quality of the spectral and spatial distortions separately, taking into account all of the downscaled bands.

$$QNR = (1 - D_\gamma)^\alpha (1 - D_s)^\beta \tag{6}$$

where α and β exponents attribute the relevance of spectral and spatial distortions to the overall quality. The two exponents jointly determine the non-linear response and have values between 0 and 1. The spectral distortion index (Dγ, Equation (7)) is defined as:

$$D_\gamma = \sqrt[p]{\frac{1}{L(L-1)} \sum_{l=1}^{L} \sum_{r=1}^{L} \left| Q(\hat{G}_l, \hat{G}_r) - Q\left( \widetilde{G}_l, \widetilde{G}_r \right) \right|^p} \tag{7}$$

where L is the number of bands to be compared (5 bands), $\hat{G}_l$ and $\widetilde{G}_l$ are the fused and low-resolution versions of band l, respectively. Exponent p is a positive integer chosen to emphasize considerable spectral differences. For p = 1 (as in this study), all the differences are equally weighted.

The spatial distortion index ($D_s$) is defined as:

$$Ds = \sqrt[q]{\frac{1}{L} \sum_{l=1}^{L} \left| Q(\hat{G}_l, P) - Q\left( \widetilde{G}_l, \widetilde{P} \right) \right|^q} \tag{8}$$

where $P$ is the high-resolution image used as a reference (band 1 at a 250 m resolution in our case), and $\widetilde{P}$ is a spatially degraded version of the $P$ image obtained by filtering with a low-pass filter. Analogously, $D_s$ is proportional to the q-norm of the difference vector, where q may be chosen to emphasize greater differences. Following the criteria used in Equation (7) (p = 1), q = 1 was used to keep all the differences equally weighted.

### 3.2.2. Spectral Mixture Analysis

The spectral mixture analysis (SMA), also known as spectral unmixing, follows the assumption that the spectral intensity in an image (or, rather, a pixel's spectral response) is created by mixtures of a limited number of surface materials, or land classes in our case. Moreover, the weighted "endmember" fractions (pure spectral signatures of a specific land-cover measured in the laboratory, in the field, or from the image itself) can be determined for each pixel [9,67]. For the SMA implementation, we used the multiple endmember spectral mixture analysis (MESMA), an advanced method of linear SMA (Equation (9)), which calculated the component fractions within a pixel, allowing endmembers to vary in type and number on a per-pixel basis [68]. For its implementation, the RStoolbox package [69] was used.

In the linear SMA, $\hat{P_{i\lambda}}$ (Equation (9)) was the spectral mixture in pixel i of wavelength $\lambda$, modeled as the sum of the N reference of endmembers, $P_{k\lambda}$, each weighted by the fraction $f_{ki}$, and the sum of these fractions must be less than or equal to 1 for each pixel. $\varepsilon_{i\lambda}$ is the residual error at $\lambda$ for the fit of N endmembers, and the respective *RMSE* (Equation (10)) is used in the optimization process to find each weighted fraction, $f_{ki}$, for each endmember:

$$\hat{P_{i\lambda}} = \sum (f_{ki} * P_{k\lambda}) + \varepsilon_{i\lambda} \tag{9}$$

$$RMSE = \sqrt{\frac{\sum(\epsilon_\lambda)^2}{B}} \tag{10}$$

where *B* is the number of bands used.

For the endmember selection of a multispectral image, this methodology allowed us to relate the number of endmembers to the number of image bands plus 1 (number of bands + 1), thus allowing for the set of equations to be solved simultaneously to create an exact solution without any error terms [9]. Given that we used the first 7 MODIS bands (previously downscaled) in this study, selecting a maximum of 8 endmembers for the algorithm was required. For this, we followed the strategy of endmember selection proposed by Painter et al. (2009) [12], where four endmembers were used for different kinds of snow (based on its grain size), rock, forest, grassland, and peatland. The ability to recognize different types of snow is provided by the sensitivity of the spectral reflectance of snow (Figure 3) to its grain size [47], which also made it possible to estimate its broadband albedo ($\alpha$) using the equation presented below:

$$\alpha = 1 - A(\theta_0)r^{B(\theta_0)} \tag{11}$$

$\alpha$ uses a relationship between the illumination angle (defined as the solar zenith angle—terrain slope) $\theta_0$ and grain size (*r*) of the snow detected in the SMA for each pixel. The constants A and B depended on the wavelength of the bands and the illumination angle [12]. We used the coefficients of all solar wavelengths since, in the SMA method, all MODIS bands were considered: A: 0.0765, 0.0648 and B: 0.2205, 0.2258 for 30° and 60° illumination angles, respectively, for short-wavelength frequencies. For the broadband albedo calculation, refer to Script N° 4 in Supplementary Material.

For the vegetation endmember representation (forest, grassland, and peatland), we created a specific seasonal endmember of vegetation classes for the study site. This avoided possible misclassifications due to seasonal changes in the spectral properties, which depended on plant phenology [70]. For this, we used different MODIS images as representatives of each season. We selected the endmembers through a supervised classification using the semi-automatic classification plugin (SCP) developed by Congedo (2016) [71] and supported by QGIS, v 3.18 [72]. Our method of the spectral mixture analysis (Script N° 4) took into account the season of the year for selecting the endmember spectral signature that needed to be used.

### 3.2.3. Spatio-Temporal Snow Reconstruction

While a significant improvement of the spatial resolution is necessary for spatial–temporal studies of snow variability in complex terrains, several confounding factors (i.e., the presence of clouds, sensor noise, and miscellaneous artifacts) made it difficult to use the snow cover algorithms directly. This problem of misclassification can be improved implementing a space–time data cube approach as a second step after the first results (SMAs), which enables a better snow-detection estimate [73,74]. For this, we followed the steps presented below.

(a)    Cloud and snow masks

The problems of differentiating snow from clouds using global MODIS products are widely known [15,19–21]. Nonetheless, these problems still persist in the current Collection v6 [16]. Since the methodology for improving the cloud mask proposed by Dozier et al. (2008) [73] was not effective in the study region, we explored an alternative method, called snow mask, based on the fSC > 0.2 detected by the previous SMA. This threshold was selected by comparing the number of snow pixels and spatial distribution with the application of other snow indices, mentioned below, on the cloud-free MODIS image from 25 September 2015. The snow mask was validated through a combination of NDSI (Equation (1)) and melt-area detection index (MADI, Equation (12)). The underlying principle was based on identifying liquid water layers that coated snow and ice in order to discriminate between frozen and liquid water [75]. The threshold proposed was a MADI $\geq$ 6, which was also used in studies of snow detection [3,76]. On the selected image, the numbers of snow pixels were SMA = 2994, NDSI = 3522, and MADI = 2927, the selected threshold (SMA 20%) being more conservative than the other analyzed thresholds (SMA at 30% = 2347 pixels and SMA at 50% = 1199 pixels).

$$\text{MADI} = \frac{\rho B_1}{\rho B_7} \tag{12}$$

This MADI expression used the reflectance ratio between the red portion of the visible ($\rho B_1$, bandwith 620–670 nm, MODIS band 1) and mid-infrared ($\rho B_7$, bandwith 2105–2155 nm, MODIS band 7) spectra.

(b)    Temporal interpolation

Considering the need to analyze the data for both space and time [77], we applied a gap-filling technique after resolving the uncertainties between the clouds and snow. This technique consisted of a time interpolation on each pixel based on the methodology described by Redpath et al. (2019) [74]. For this, we used a time window of 15 days (i.e., the results were saved on the 7th day) with a maximum of 5 days for consecutive cloud conditions. A larger cloudy period of consecutive days (e.g., 10) did not have a good performance, and thus appeared as missing data. For temporal interpolations in each pixel, we used a smoothing cubic spline interpolation [78] that was also used in the work of Dozier et al. (2008) [73]. The fractional snow cover and broadband albedo interpolations were implemented separately using the snow mask previously defined. Interpolated values out of range were changed by its limits (0 and 1). The gap-filled time series of snow properties provided the basis for the further analysis of snow-cover climatology and variability. For the implementation, we used both raster [61] and space–time [79] R packages.

The scripts (5) and MODIS products referred to in the previous sections are attached as part of the Supplementary Materials.

### 3.2.4. Ground Validation

To validate the spatial–temporal reconstruction of snow properties, we used the daily values of albedo and snow height measurements at AWS Tres Morros averaged over 3 h at

10:00, 11:00, and 12:00 h local time, considering that MODIS Terra passed over the study area at around 11:00 h local time.

As for the quality assessment of the meteorological data, there were conflicting criteria for its implementation. On the one hand, some authors indicated that "erroneous" data could alter the analysis of the time series. On the other hand, other authors believed that the implementation of quality control methods could exclude valid values [80], often dismissing extreme values necessary in the climate analysis [81]. In this work, we only reviewed the existence of anomalous values, but no additional statistical tests were implemented for the data validation or confidence ranges regarding the exclusion of extreme values.

To validate the proposed methodology, we first compared the snow height (Snow_h) and albedo (Snow_al) measurements from AWS Tres Morros and the snow property reconstruction, based on the steps described above, for the respective pixel and the days where both coincided. The values considered for comparison were the percentage (%) of snow detected (higher than 5% in fSC) based on the AWS measurements, considering days with a snow height higher than 5 cm based on a vegetation height below the snow sensor, as in the description presented by Rodell and Houser (2004) [82].

We used Pearson's correlation and RMSE, exploring the linear and non-linear relationships between the variables. For the non-linear case, we used the generalized additive models (GAM) approach using the cubic regression spline as a smoothing function [83]. The GAM allowed for the detection of non-linear relationships between the response and multiple explanatory variables, also called additive models, fitting a smoothing curve through the data [60,83]. The AIC and deviance explained (as a measure of the quality of fit GAM models and for its Gaussian family, as in our case, was equal to the variance explained) were also used for model evaluations.

### 3.2.5. Snow Cover Variability

A minimum threshold of an fSC > 20% was used to consider a pixel as snow-covered, which was different from the 50% proposed by Redpath et al. (2019) [74] due to the high interaction between the snow and vegetation in the study area. In the pixel corresponding to AWS Tres Morros, we had a combination of peatland, Nothofagus forest, shrubs, and grasses. The daily and monthly temporal variabilities of the snow-covered area and the percentage of the study area (referred to the 4 watersheds) covered by snow was used to determine the seasonal snow duration in the cold season (April–October). The number of snow days were also considered as a spatial indicator of snow persistence in the Brunswick Peninsula.

To study the snow cover variability and trend analysis, we used Mann–Kendall [84] and Sen's slope [85] non-parametric methods, widely applied in climate trends analyses [86]. Before applying these trend analyses, we implemented an exponential filter [87], specifically, in our case, the Holt–Winters filter, which considered both seasonal and non-seasonal behaviors that depended on the time series analyzed, and also minimized the square prediction error [88]. We also used the time-series decomposition additive approach [89]. To complement the temporal analysis that considered the overall trends over the entire study area, we calculated trends of each pixel considering the variation in the number of snow days in the time series of annual snow days. With both products, covering spatial and temporal variabilities, we explored the relationships that could explain the spatial distribution of the trends detected. For this, we used the PCA and further linear and non-linear relationships with spatial variables, such as the latitude, longitude, exposure orientation (aspect), and height. The R packages used were wql [90], stats [88], and mgcv [83].

## 4. Results

### 4.1. Spectral Fusion

The comparative study between different spectral fusion methods followed the methodology described by Wang et al. (2015) [55]. Linear regression results, following (Equation (3)),

were reviewed using both snow and snowless scenes over a selected area in the Brunswick Peninsula. Data from MODIS band 5 were also included, not initially considered in Wang et al. (2015) [55]. Table 2 shows a good linear fit in snowless areas between band 1 (as an explanatory variable) and bands 3, 4, 5, 6, and 7, as also indicated by Wang et al. (2015) [55]. However, it is necessary to re-evaluate this relationship in snow-covered areas since the linear fit is less convincing over snow conditions. Figure 4 shows the reflectance values of MODIS at a 500-m spatial resolution (MOD09GA) in both snow and snow-free conditions.

**Table 2.** The results of the OLS regression proposed by Wang et al. (2015) in the Brunswick Peninsula.

| | **Snow Image** | **Snowless Image** |
|---|---|---|
| $Band_i$ | $Band_i \sim Band_1$ | $Band_i \sim Band_1$ |
| $Band_3$ | $R^2 = 0.96$<br>RMSE = 0.229 | $R^2 = 0.91$<br>RMSE = 0.066 |
| $Band_4$ | $R^2 = 0.99$<br>RMSE = 0.094 | $R^2 = 0.84$<br>RMSE = 0.074 |
| $Band_5$ | $R^2 = 0.39$<br>RMSE = 0.264 | $R^2 = 0.22$<br>RMSE = 0.165 |
| $Band_6$ | $R^2 = 0.54$<br>RMSE = 0.229 | $R^2 = 0.73$<br>RMSE = 0.133 |
| $Band_7$ | $R^2 = 0.41$<br>RMSE = 0.409 | $R^2 = 0.93$<br>RMSE = 0.075 |

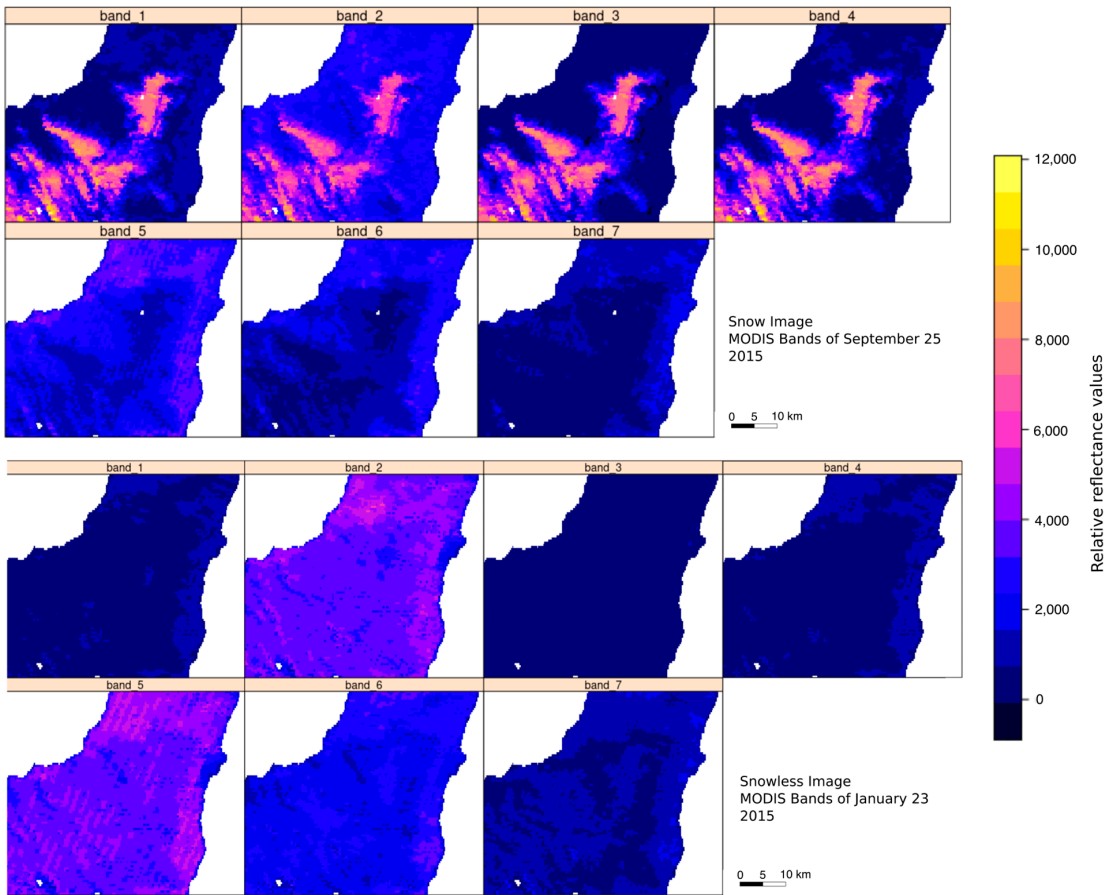

**Figure 4.** Reflectance values of MODIS (MOD09GA) bands in snow and snowless images. The reflectance values ([0, 1]) are stretched to match the MODIS reflectance scale ([−100, 16,000]).

### 4.1.1. The First Term of Spectral Fusion: The Linear Relationship

As a first step and given the different results for the snow and snowless images, using the linear relationship proposed by Wang et al. (2015), coarse band i ~ band 1 (band i in range 3 to 7), we explored other relationships and achieved better results than the relationships detailed in Table 3.

**Table 3.** Downscaling using spectral fusion, showing the first term (Equation (2)) as a new regression relationship based on coarse MODIS bands 3 to 7 at 500 m, using independent variables as the coarse-band 1 and the resampled DEM at a 500 m resolution (DEM$_{500}$).

| Coarse Band | New Relationship Proposed | OLS Regression Results |
|---|---|---|
| Band 3 | $Band_3 \sim Band_1$ | $R^2 = 0.96$<br>RMSE = 0.230 |
| Band 4 | $Band_4 \sim Band_1$ | $R^2 = 0.99$<br>RMSE = 0.094 |
| Band 5 | $Band_5 \sim Band_1 + DEM_{500}$ | $R^2 = 0.52$<br>RMSE = 0.234 |
| Band 6 | $Band_6 \sim Band_1 + DEM_{500}$ | $R^2 = 0.68$<br>RMSE = 0.319 |
| Band 7 | $Band_7 \sim Band_1 + DEM_{500}$ | $R^2 = 0.61$<br>RMSE = 0.333 |

An exploratory analysis of the different MODIS bands over a reference MODIS image on 25 September 2015 showed the need for normalization by a logarithmic transformation since the reflectance values in each band had an exponential distribution [60]. After this normalization, as a first stage, an OLS adjustment was performed using, as an independent variable, MODIS band 1 and the resampled DEM at a 500-m spatial resolution (DEM$_{500}$), as indicated in Table 3. The distribution of the residuals shows a clear violation of heterogeneity, justifying the recommendation of Wang et al. (2015) [55] to use a GLS instead of an OLS model. Additionally, the existence of a spatial correlation was reviewed with a variogram analysis of its residuals (Figure 5), with a clear spatial dependence that was considered.

Using the new relationships shown in Table 3, a GLS model was implemented for the coarse bands (GLS$_{500}$), with a selection of variance structures for each band, according to the protocol, as indicated by Zuur et al. (2009) [60]. For the example image (25 September 2015), the following variance structures were selected: Band\_3$_{500}$ power of the covariate variance structure; Band\_4$_{500}$ exponential variance structure; and Band\_5$_{500}$, Band\_6$_{500}$, and Band\_7$_{500}$ constant plus-power values of the variance covariate function.

The GLS model performed better than the OLS model according to the AIC, but not compared to the results obtained with a GWR$_{500}$. The GWR model has the advantage of generating coefficients for each pixel. The coefficients of determination ($R^2$) between the modeled coarse bands using GWR$_{500}$ and the original coarse bands varied between 0.97 and 0.99 were, Band\_3$_{500}$ = 0.99, Band\_4$_{500}$ = 0.99, Band\_5$_{500}$ = 0.97, Band\_6$_{500}$ = 0.98, and Band\_7$_{500}$ = 0.97, with low RMSE values for all models (Table 4).

Finally, to obtain the first term of the spectral fusion ($Z_{v1}^l$) of Equation (2), we needed to use these relationships (at coarse resolutions) and implement them at a higher resolution (250 m). For the cases of OLS$_{500}$ and GLS$_{500}$, the key issue was the estimation of the regression coefficients for the coarse bands, assumed to be universal at different spatial resolutions. Consequently, the relationship built at coarse spatial resolutions could be applied at a higher spatial resolution [56]. This assumption was the same for the GWR$_{500}$ model; but, since there were coefficients for each pixel ($\beta_0$ and $\beta i$ in Equation 4), these were assigned to the highest-resolution imagery on a pixel by pixel basis of the original data as an additional information layer.

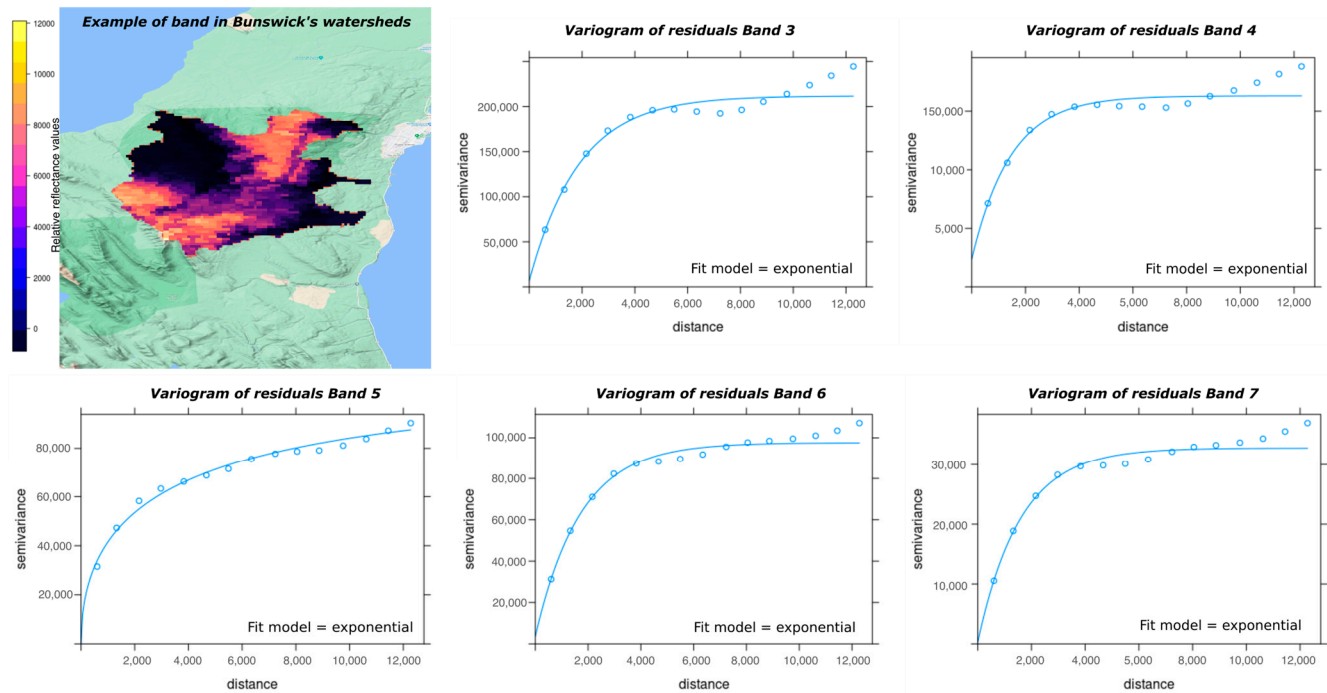

**Figure 5.** Variograms of linear regression residuals for each coarse band in the Brunswick Peninsula watersheds. The colors in the upper-left image represent the reflectance values (yellow represents high and black represents low values) of band 1 in an image from 25 September 2015.

**Table 4.** Models and downscaling results for subset selections in the snow-covered-image areas in the Brunswick Peninsula. All indexes are dimensionless: spatial distortion index (Ds), spectral distortion index (D$\gamma$), quality index without reference (QNR index), and universal quality index (UQI).

| | OLS$_{500}$ | GLS$_{500}$ | GWR$_{500}$ | OLS_ATAK$_{250}$ | GLS_ATAK$_{250}$ | GWR_ATAK$_{250}$ |
|---|---|---|---|---|---|---|
| Band_3$_{500}$ | $R^2 = 0.97$<br>RMSE = 0.230<br>AIC = $-175$ | $R^2 = 0.97$<br>RMSE = 0.372<br>AIC = $-2877$ | $R^2 = 0.99$<br>RMSE = 0.110<br>AIC = $-2500$ | UQI = 0.92 | UQI = 0.94 | UQI = 0.93 |
| Band_4$_{500}$ | $R^2 = 0.993$<br>RMSE = 0.085<br>AIC = $-3713$ | $R^2 = 0.993$<br>RMSE = 0.101<br>AIC = $-6053$ | $R^2 = 0.99$<br>RMSE = 0.038<br>AIC = $-6296$ | UQI = 0.93 | UQI = 0.93 | UQI = 0.93 |
| Band_5$_{500}$ | $R^2 = 0.64$<br>RMSE = 0.152<br>AIC = $-1654$ | $R^2 = 0.63$<br>RMSE = 0.153<br>AIC = $-1727$ | $R^2 = 0.97$<br>RMSE = 0.047<br>AIC = $-5404$ | UQI = 0.89 | UQI = 0.98 | UQI = 0.89 |
| Band_6$_{500}$ | $R^2 = 0.69$<br>RMSE = 0.304<br>AIC = 822 | $R^2 = 0.65$<br>RMSE = 0.392<br>AIC = 222 | $R^2 = 0.98$<br>RMSE = 0.074<br>AIC = $-3838$ | UQI = 0.96 | UQI = 0.99 | UQI = 0.91 |
| Band_7$_{500}$ | $R^2 = 0.58$<br>RMSE = 0.302<br>AIC = 794 | $R^2 = 0.53$<br>RMSE = 0.359<br>AIC = 498 | $R^2 = 0.97$<br>RMSE = 0.080<br>AIC = $-3545$ | UQI = 0.97 | UQI = 0.99 | UQI = 0.89 |
| | | | Ds | 0.042 | 0.043 | 0.039 |
| | | | D$\gamma$ | 0.015 | 0.009 | 0.016 |
| | | | QNR Index | 0.943 | 0.949 | 0.946 |

### 4.1.2. The Second Term of Spectral Fusion: The Kriging Interpolation

The next step was to apply the ATAK interpolation to residuals of the coarse regression (500 m) to each model (OLS$_{500}$, GLS$_{500}$, and GWR$_{500}$), so as to redistribute the error at a resolution of 250 m. First, we needed to select the best model representing the residuals' spatial

correlations (variograms). Once this model was estimated for each band, we proceeded with implementing the ATAK interpolation to obtain the second term of Equation (2) ($Z_{v2}^l$). This comparative process is detailed in Script N°1 of (Supplementary Materials and codes), and its results are shown in Table 4.

While in the coarse-resolution data, the results of the $GWS_{500}$ model were the best option; once implemented in the fine-resolution data (250 m), the three ATAK models achieved a similar behavior based on the QNR index. The best results correspond to the $GLS\_ATAK_{250}$ model, with a QNR index of 0.949. However, regarding the results for each band's UQIs, the GLS model was the best (Table 4). The QNR index represents the performance of each spectral fusion model, where, in our case, we considered the spatial (Ds) and spectral (Dγ) distortions with equal relevance. An experiment with the $GWR_{250}$ model without an ATAK interpolation was performed, obtaining a QNR index of 0.914, which increased to 0.946 after applying the ATAK interpolation ($GWR\_ATAK_{250}$). Obviously, the improvement was not so significant compared to the OLS and GLS models. This can be explained in part by the spatial correlations of its residuals (Equation (2)), which lacks a good model representation. On the other hand, the computation time of the $GLS\_ATAK_{250}$ method was almost three-times faster than the $GWS\_ATAK_{250}$, which was preferable, given that one of our goals was to have an operational methodology for a large amount of data to be processed.

All algorithms were contained in Script N°2 (in Supplementary Material), which included (i) the selection of the best variance structure of the model for each band, (ii) the selection of the best variogram model of the residuals, and (iii) controls on the possible issues in this downscaling batch process, managing the missing data, outliers, and image data problems (we included a threshold for detecting a lack of spectral information (for saturation or other spectral problems) of a standard deviation of the digital numbers lower than 10, considering the whole picture, based on our experiments). Figure 6 shows the residual maps of the GLS regression at 250 m and the sum of these two products (Equation (2)), compared with the corresponding coarse band.

*4.2. SMA*

Before implementing the SMA, as was mentioned in the methodology, it was necessary to find the seasonal spectral signatures (endmembers) of the principal vegetation types present in the Brunswick Peninsula. In an explorative analysis of the spectral signature and its representation as endmembers in the Brunswick Peninsula, we selected forest, grassland, and peatland, in addition to the 4 categories of snow and bare ground. Table 5 shows the days selected to detect the seasonal spectral signatures for each vegetation type.

**Table 5.** Image selection for seasonal endmember extraction in the Brunswick Peninsula.

| Southern Hemisphere Season | Image Date |
|---|---|
| Summer | 17 January 2015<br>16 January 2016 |
| Autumn | 29 April 2016<br>5 May 2016 |
| Winter | 3 and 11 September 2016 |
| Spring | 16 and 18 October 2016 |

Figure 7 shows how each spectral signature varies seasonally. Forest presents a rather low reflectance compared to grassland and peatland. This seasonal variability reaffirms the importance of considering seasonal changes in the vegetational endmembers for this type of spectral classification.

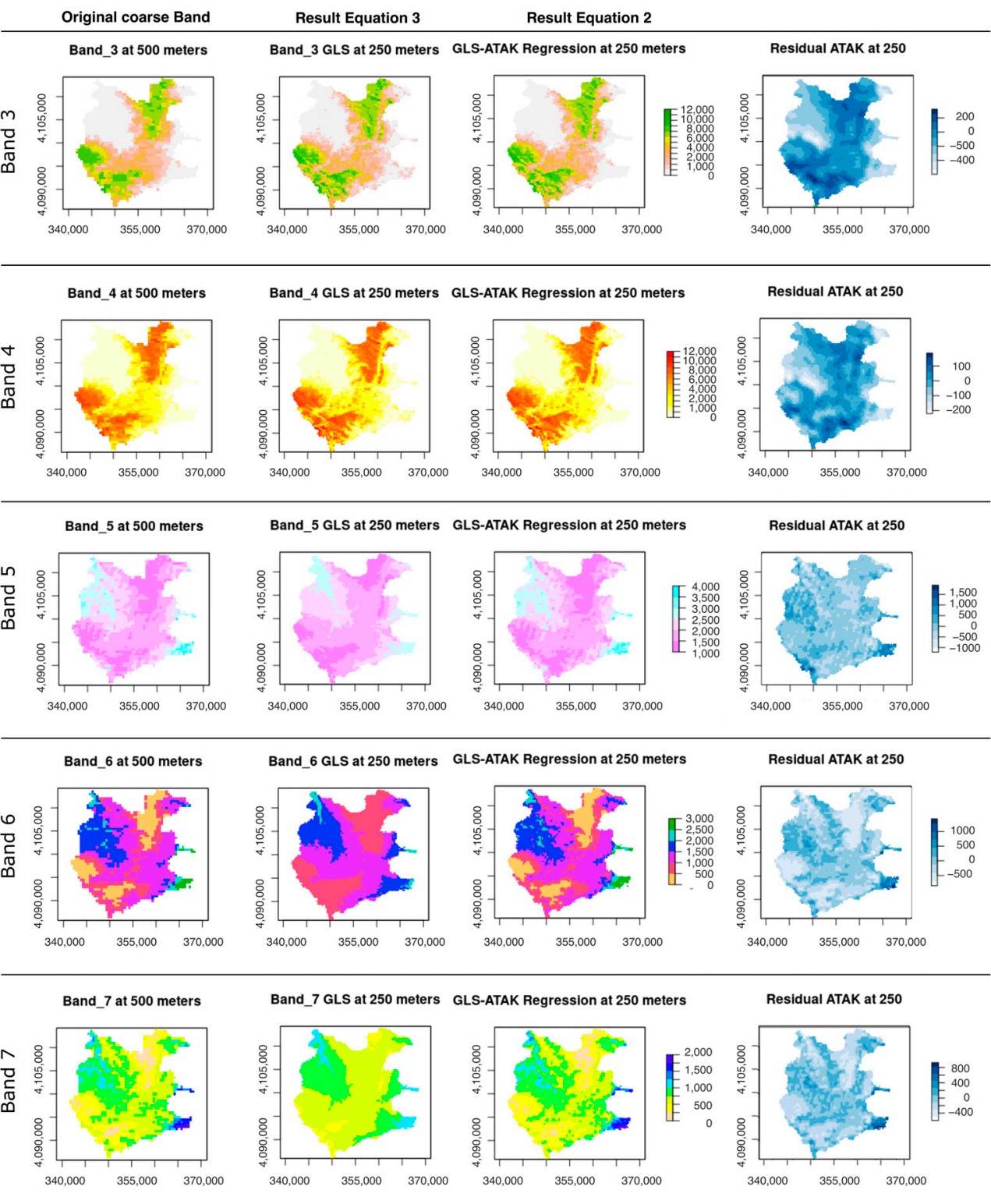

**Figure 6.** Spectral fusion results using the GLS-ATAK models. From left to right: residuals of ATAK at 250 m, GLS at 250 m, GLA-ATAK at 250 m, and coarse band at 500 m. The 16-bit reflectance units for each band are expressed at a scale of [−100, 16,000], and the coordinates of the map are in UTM. MODIS image of 25 September 2015.

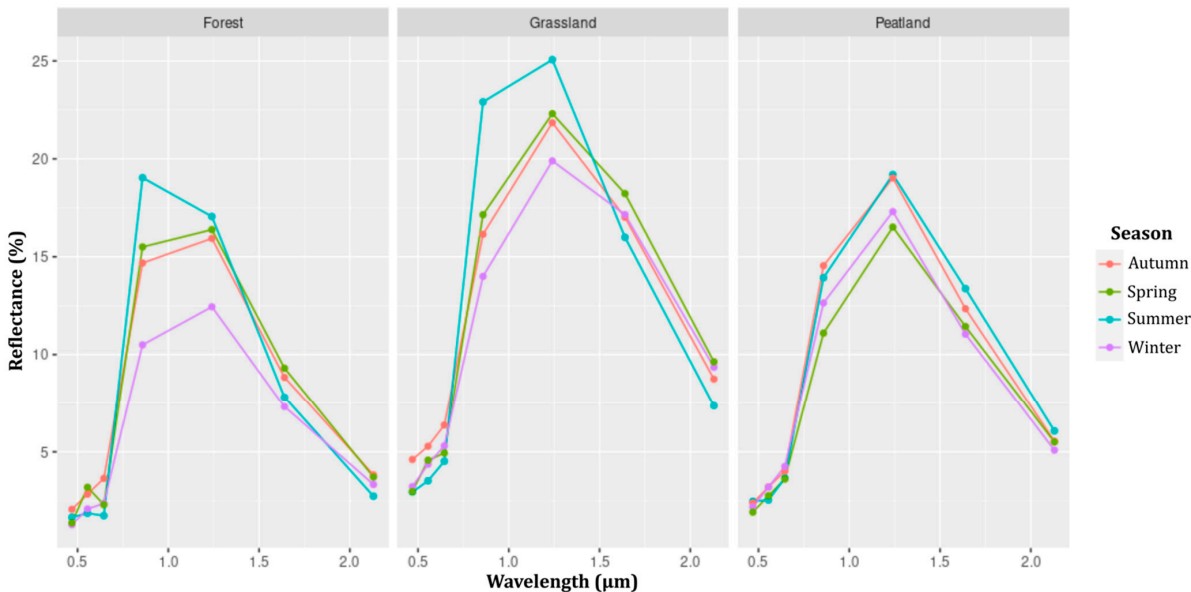

**Figure 7.** Endmember signatures and their seasonal variations in the Brunswick Peninsula. Reflectances are in % and represent spectral albedos for each MODIS band.

In Figure 8, the results of the SMA are presented. The snow fraction (Figure 8b) is a sum of the different snow categories (based on grain size; Figure 8d) used for each pixel. This categorization was used for the calculation of broadband albedo values (Figure 8c). The elevation model (Figure 8a) is shown because the albedo is a function of the illumination angle, which in turn is affected by the surface topography. As is mentioned in the Materials and Methods Section 3.2, the processing shown in this figure is performed automatically for each MODIS image at a 250 m resolution (Script N° 4 of Supplementary Material).

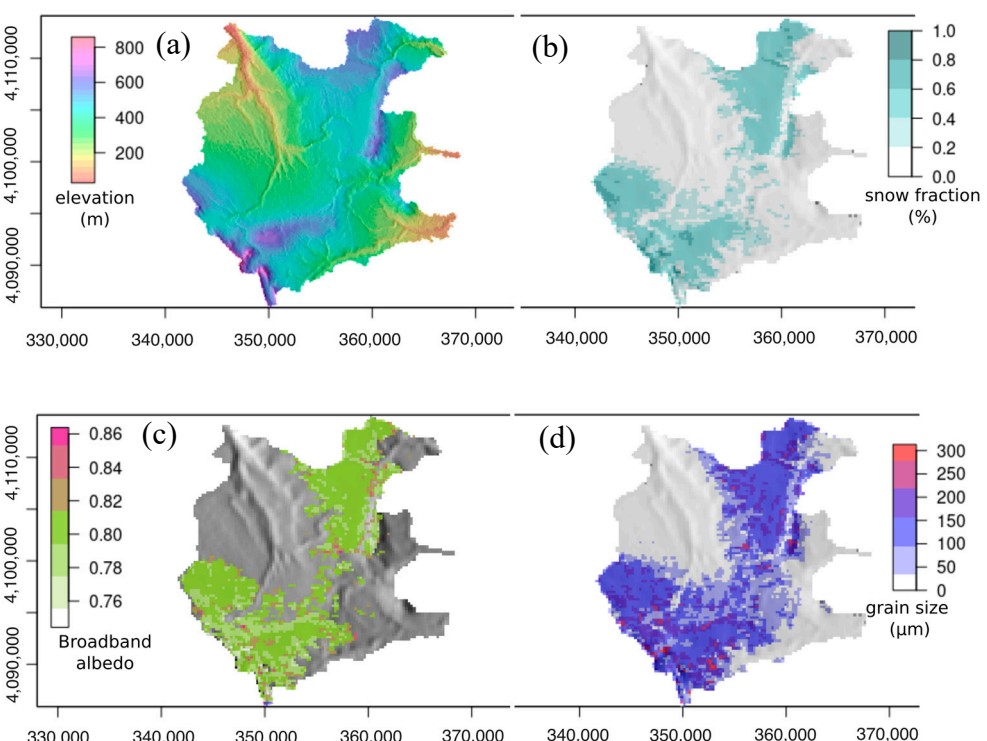

**Figure 8.** Spectral mixture analysis (SMA): (**a**) elevation model; (**b**) fractional snow cover for each pixel; (**c**) broadband albedo; and (**d**) average grain size of snow for each pixel (images for B, C, and D dating from 25 September 2015).

### 4.3. Spatio-Temporal Snow Reconstruction

4.3.1. Cloud and Snow Masks

After performing the SMA, we contrasted the results of fractional snow cover, grain size, and cloud cover mask over 3 days with different atmospheric conditions. The cloud mask confused snow and cloud cover in all conditions, limiting its use (Figure 9). Considering that the west side of southernmost Patagonia is affected by a high number of cloudy (77–86%) and precipitation (81–88%) days throughout the year [44], this issue must be considered, especially in an area of climatic transition as occurs in the Brunswick Peninsula. The RGB compositions of the grain size (red), NDSI (green), and MADI (blue) showed a good discrimination of snow and cloud cover. A snow mask was generated by applying the thresholds indicated in the Materials and Methods Section. Using this snow mask to discriminate the snow pixels (value 1 as snow, value 0 as no snow, and nan as indeterminate), we improved the discrimination between the clouds and snow, and applied it to reconstruct the snow fraction and broadband albedo. Script N° 4 of Supplementary Material describes this process.

4.3.2. Temporal Interpolation

Taking advantage of the new snow mask proposal, developed in the previous section, and its implementation based on 21 years of MODIS data in the study area, we performed a temporal interpolation in each pixel for the fSC and albedo. Figure 10 presents how the cubic spline smoothing function, described in detail in the Materials and Methods Section, works as a moving window of 15 days (7 days forward and 7 days backward) and can fill the missing data for fractional snow cover and albedo based on past and present behaviors for each pixel due to cloud cover conditions. We can note that, after the implementation of the interpolation technique, several grid points with missing values have been filled, which is particularly noticeable on 6 and 8 August because they have several problems with cloud interference, as we can see in Figure 9. This process was also implemented for batch processing (Script N° 5 in the Supplementary Material).

### 4.4. Ground Validation with AWS Data

Figure 11 shows the annual winter variability of the snow height measured at AWS Tres Morros from July 2018 (installation date) to December 2020, also showing when the snow height exceeds 20 cm for the sampling period. The data were processed and compared with the MODIS reconstructed data corresponding to the 250 m pixel location. For the AWS albedo data from Tres Morros, a minimum snow height of 20 cm was established to consider representativeness, avoiding problems with mixed snow and vegetation signals. The valid albedo data were limited to July (only four days), August, and September of 2018 due to problems with the field data, with mean (maximum) values in August of 0.82 (0.96) and September 0.72 (0.87). We only found 34 of 159 valid AWS albedo measurements (with a snow height > 20 cm) that coincided on the same day with the reconstructed MODIS albedo at a 250 m resolution, exhibiting a poor correlation (Pearson's correlation of 0.1).

For the fSC reconstruction (SF_Rec) from the MODIS imagery in 2018–2020, 125 days of MODIS data at the Tres Morros pixel were compared with AWS Tres Morros snow height measurements. A Pearson's correlation of 0.45 was found in the linear relation (SF_Rec ~ Snow_h) and RMSE of 0.105. However, exploring the GAM model (SF_Rec ~ s(Snow_h)), the RMSE reduced to 0.083 and $R^2$ increased to 0.48. AIC values were -63 (linear) and -249 (GAM), respectively, indicating that the GAM model provided the preferred approach. Nonetheless, for snow height values below 20 cm (presented in the relation GAM 20 cm), fSC values of 0 were found, suggesting that the threshold of the fSC in the snow mask (fSC > 0.2) for this pixel left some fSCs undetected. Table 6 shows the results of the AWS Tres Morros data vs. the reconstructed values. As for the confidence of snow detection at this pixel, we found a 98% coincidence between the fSC (with a threshold of 10%) and snow height measurement at AWS Tres Morros, using a threshold of 5 cm.

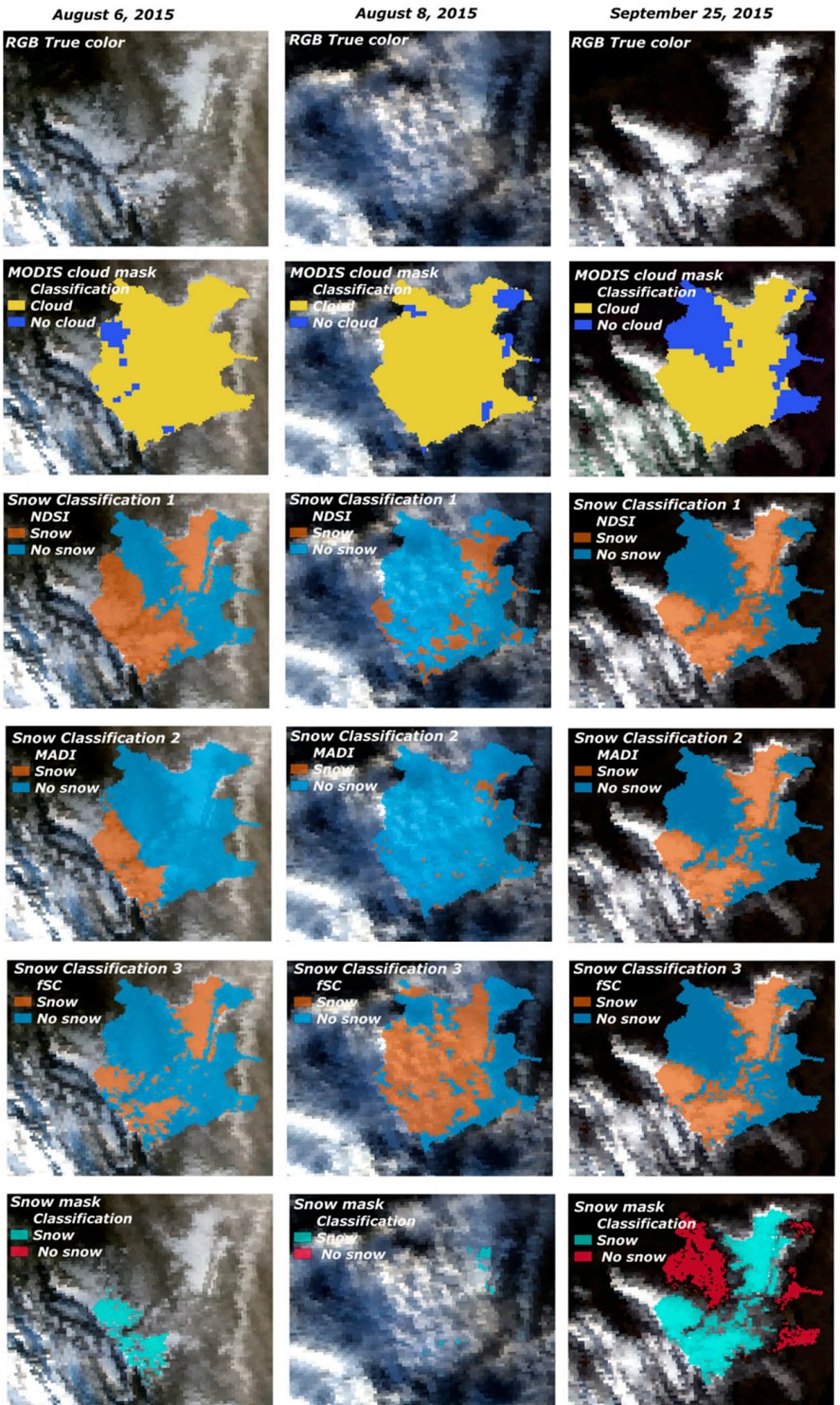

**Figure 9.** Snow and cloud cover and new snow classifications for three types of cloud cover: (i) presence of both snow and clouds, including mist (6 August 2015), (ii) mostly cloudy (8 August 2015), and (iii) snow with clear skies (25 September 2015). The top row shows RGB true-color images; second row, cloud mask product of MODIS (MOD35_L2); third row, first snow classification based on NDSI (snow threshold $\geq$ 0.4); fourth row, second snow classification based on MADI (snow threshold $\geq$ 6); fifth row, third snow classification based on fractional snow cover fSC (snow threshold $\geq$ 0.2); bottom row, snow mask generated based on the condition that all three snow classification indexes must agree.

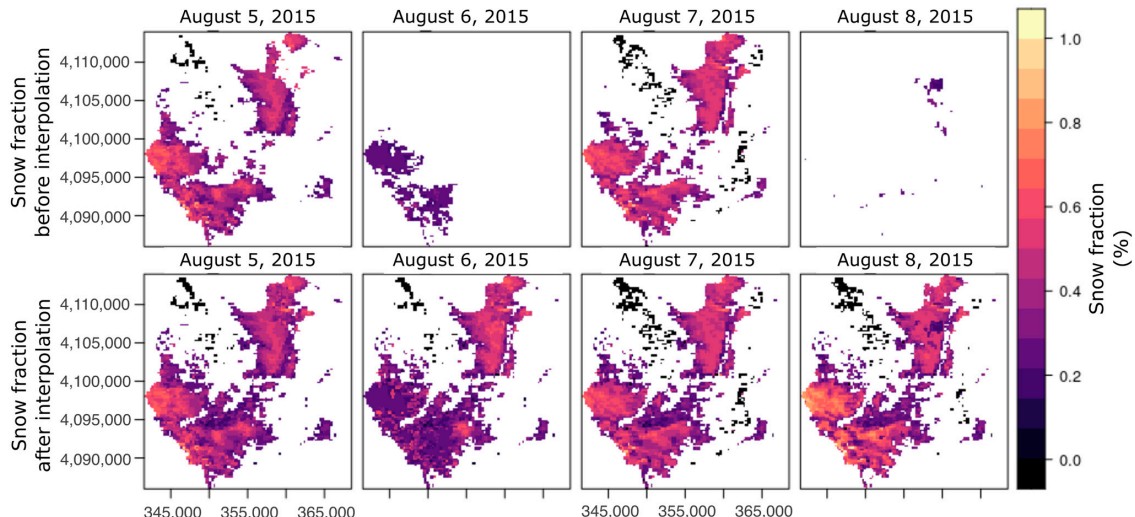

**Figure 10.** Snow fraction (fSC) spatial–temporal reconstruction, before (**top**) and after the interpolation (**bottom**).

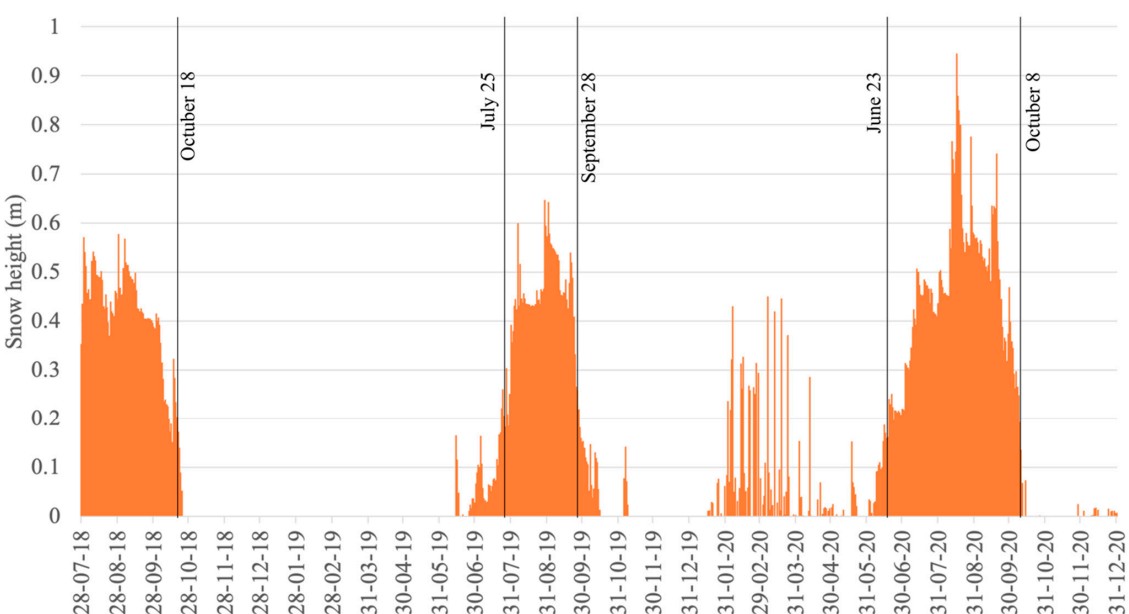

**Figure 11.** Snow height measurements at AWS Tres Morros. Each black line represents the time when the snow height decreases to below 20 cm.

**Table 6.** AWS Tres Morros snow height measurements (Snow_h) vs. fSC reconstruction from MODIS imagery (SF_Rec) in 2018–2020.

| Relations Evaluated | |
| --- | --- |
| **Linear**<br>SF_Rec ~ Snow_h | $R^2 = 0.20$<br>RMSE = 0.106<br>AIC = $-63$ |
| **GAM**<br>SF_Rec ~ s (Snow_h) | $R^2 = 0.45$<br>RMSE = 0.083<br>AIC = $-249$ |
| **GAM 20 cm**<br>SF_Rec ~ s (Snow_h) | $R^2 = 0.05$<br>RMSE = 0.067<br>AIC = $-251$ |

*4.5. Reconstructed Snow Cover Variability in the Brunswick Peninsula*

Figure 12a shows a box plot of the daily snow cover area during the cold season for each year for the entire study area. The highest number of outlier values coincides with the years that present a short snow season (2003, 2014, 2016, and 2017); although, during those years, several days show high levels of snow. The snow season (shown in Figure 12b,c) was defined as a minimum percentage of snow cover area of 10% (Brunswick Peninsula), which was highly related to the number of skiing days at the ski center located in Cerro Mirador (Pearson's correlation of 0.9, RMSE of 14 days, [3]), showing an average of 68 days (Figure 12c) for the 2000–2020 period. For the last 9 years (2012–2020), 7 years were below this average. The monthly trends analysis (Figure S1 in Supplementary Materials) of both the 2000–2020 and 2010–2022 periods, using the MK and Sen's slope, did not show a significant trend (at 95%). However, the monthly behavior of the time series changed during the 2010–2020 period, with September showing a decreasing trend of 7.5 km$^2$ per year, and after applying a Holt–Winters filter, it showed a significant (at 95%) decreasing trend of 4.7 km$^2$ per year. The reconstructed snow days (snow season days) for the whole period (2000–2020) showed a trend of +0.11 days/year (not significant); but, by applying an exponential filter, it presented a significant trend of +0.54 days/year in the Brunswick Peninsula ($p$-value < 0.005). However, for the last 10 years (2010–2020), a significant decreasing trend of −4.64 snow days/year was observed in the Brunswick Peninsula ($p$-value < 0.001).

Figure 13a identifies the spatial distribution of the snow season duration in days (as an average in 2000–2020) for each pixel, showing 65 and 75 snow days in Cerro Mirador and Tres Morros, respectively. Figure 13b shows the significant trend of yearly snow days (snow days/year) of the whole series, with a trend of −0.5 snow days/year (decrease) and +0.3 snow days/year (increase) in Cerro Mirador and Tres Morros, respectively. The yearly snow-days dataset (Figure 13a,c) is correlated with terrain information (elevation and aspect) and spatial location (latitude and longitude); but, these correlations are not so representative in the yearly snow-days trend (Figure 13b,d). The PCA analysis (presented as supplementary information, Figure S2) for the model of average snow days (Mod_av) identified that all terrain and spatial variables had a significant contribution, explaining 54.9% of the variance using the two principal eigenvalues. This is also indicated by the $p$-value (less than 0.05) of all terms in the linear regression, with the average annual snow season duration data normalized by a logarithmic function, given the exponential distribution of its data. For the case of the model of the snow-days trend (Mod_trend), all terrain and spatial variables made a significant contribution, which explained 56% of the variance based on the two principal eigenvalues, which was also identified in the linear model with a significance ($p$-value) < 0.05. For Mod_av, the elevation variable yielded the highest Pearson's correlation (−0.63) for a linear relationship, with Mod_trend elevations of longitude (long) and latitude (lat) of 0.11, 0.15, and 0.14. In both cases (Mod_av and Mod_trend), a GAM was implemented using a cubic spline regression as a smoothing function (Wood, 2017; Zuur et al., 2009). Figure 13a (Mod_av) and Figure 13b (Mod_trend) present the elevation components of the respective GAMs, which shows the highest correlation with the response variables in both cases. Elevation has the most pronounced effect on snow cover (snow days/year) in the study area with a range of 380–500 m a.s.l for yearly snow days. However, in the case of yearly trends of snow days, elevation does not have a strong correlation. The implemented GAMs explain 64.7% (for snow days) and 22.7% (for yearly trend) of the total variance, respectively.

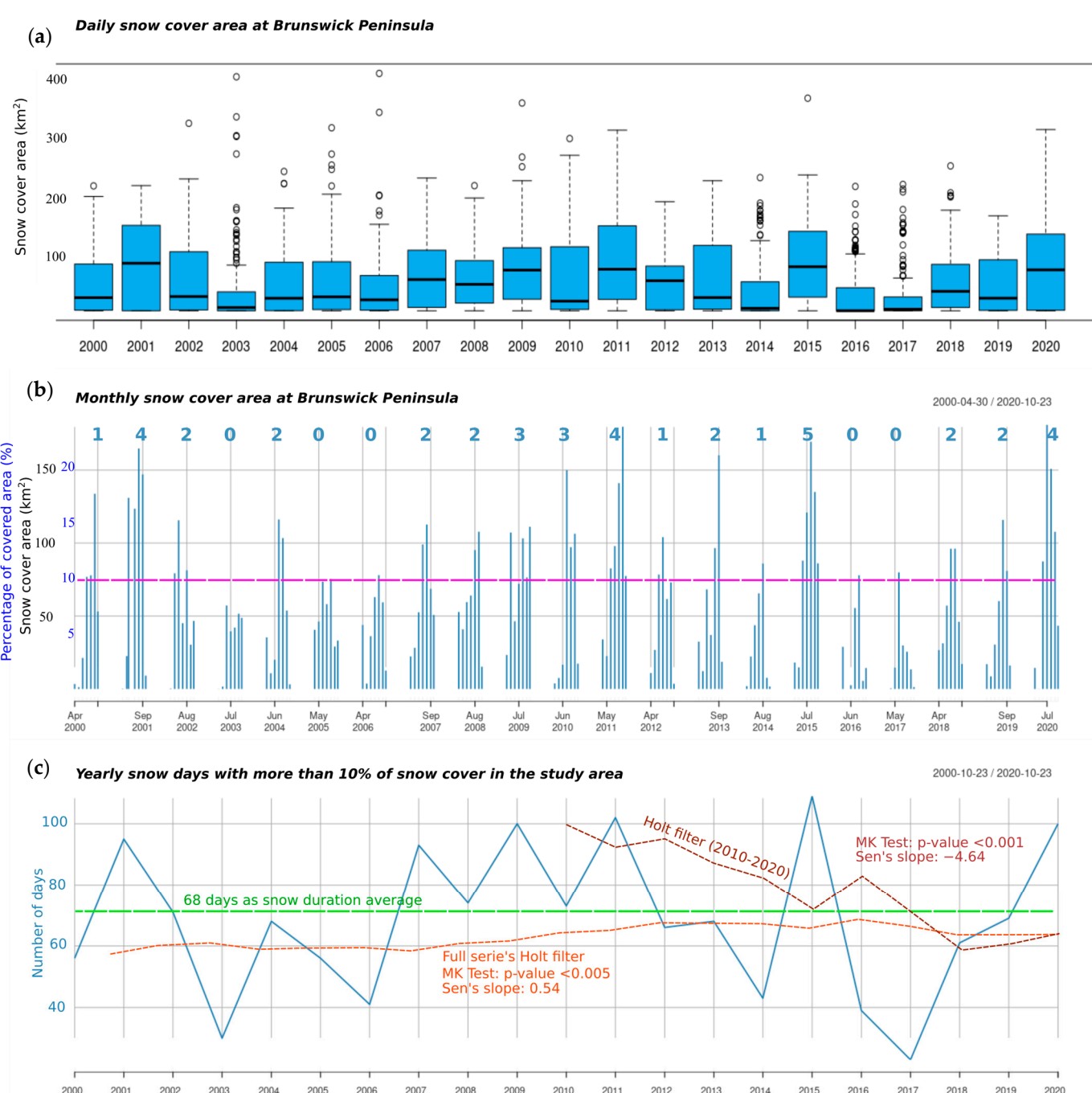

**Figure 12.** Snow cover variability in the Brunswick Peninsula from MODIS reconstruction for the entire study area. (**a**) Daily snow cover area by year as boxplots, with empty circles representing outliers. (**b**) Monthly snow cover area as both km² and percentage. The horizontal pink line represents 10% of the total study area as a definition of the snow season and the numbers in blue above represent the number of months during the snow season. (**c**) The blue line represents the number of days per year that the snow cover exceeds 10% of the total study area. The orange dashed line represents the exponential filter of the number of days per year for the whole series, which shows a significant trend of +0.54 days/year. The red dashed line represents the exponential filter of the number of days per year for the 2010–2020 period, showing a significant decreasing trend of −4.64 day/year. The green line presents the average of the snow season days in 2000–2020.

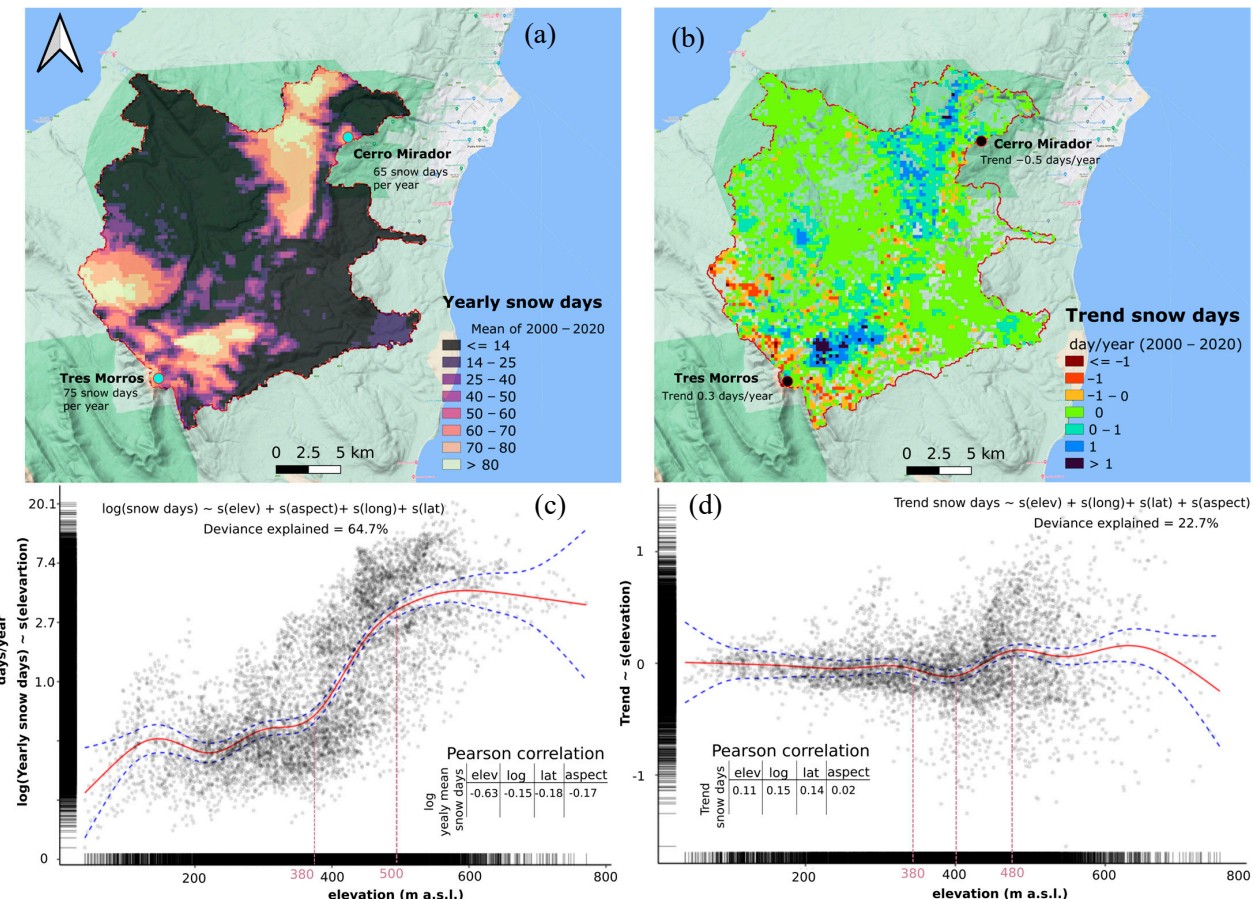

**Figure 13.** Spatial and temporal variabilities of snow cover in the Brunswick Peninsula from the MODIS reconstruction. (**a**) Spatial variability of snow days per year as an average for the period 2000–2020; (**b**) yearly significant trend of reduction in snow days for each pixel (in days/year). (**c**,**d**) Show the elevation component of the generalized additive model (GAM). The red line is a spline fit, with confidence intervals (at 95%) shown as blue dashed lines. (**c**) Model showing the altitudinal distribution of snow days as an average for the period 2000–2020 and (**d**) model showing the altitudinal distribution of yearly trend of snow days.

## 5. Discussion

### 5.1. Method Improvement

The methods presented here include tests of recent downscaling methodologies [55,59] regarding both the accuracy and speed of implementation (computing cost) to improve the spatial resolution of wintertime acquisitions, since these procedures have mainly been performed in summer conditions [55]. The method proposed here (shown in Figure 1) is evaluated for each band (UQI), as well as using a global analysis (QNR index) for the preservation of spectral ($D_\gamma$) and spatial ($D_s$) characteristics. An improvement was achieved for all indices, and also regarding the computational implementation. It is important to consider that our method was compared for two images, one image with snow and the other without snow, in contrast to the work of Wang et al. (2015) [55], where these comparisons were performed on a single and clear image (without clouds and without snow). It would be interesting for future applications of this methodology to create a longer time series to provide a more extensive assessment of the images processed. However, this was outside our main scope since we concentrated here on describing and implementing this new method in southern Patagonia.

Of high importance are cloud and snow discriminations. After testing different approaches between cloud and snow discriminations (i.e., [73,74,91]), we proposed a new method. The proposed method combined SMA, NDSI, and MADI using snow-detection

thresholds, semi-automatically processing 21 years of MODIS images, which allowed us to avoid the usual confusion between snow and cloud classifications, and obtaining daily values of snow fSCs and albedo values at a spatial resolution of 250 m.

The proposed methodological validation includes (1) the spectral fusion for downscaling process, (2) the spectral mixture analysis, (3) the snow/cloud discrimination using a snow mask, and (4) the temporal interpolation. These steps were compared with ground measurements from AWS Tres Morros. Our reconstruction showed a coincidence of 98% between the fSC (with a threshold fSC $\geq$ 10%) and the measurements of snow height (Snow_h) at the location of AWS Tres Morros (with a threshold Snow_h $\geq$ 5 cm). We also observed a relationship (generalized additive model) between SF_Rec ~ s (Snow_h) that explained a variance of 55 %. However, this relationship depended on the height of vegetation; so, it could not be extrapolated to other pixels but may have been extrapolated to the past for the pixels tested. Out of a total of 159 days of valid field AWS albedo measurements, only a period of 34 days overlapped with the satellite reconstructed data, exhibiting a poor correlation (Pearson's correlation of 0.1). AWS Tres Morros is located below the tree line; so, the pixel used for this validation had a relevant vegetation component, with an fSC of 20% to 50% and a maximum snow height of 80 cm. It is necessary to measure the albedo in the field at several locations within 250 $\times$ 250 m pixels to obtain an average representation of albedos to obtain a better correlation between the field and satellite data.

Regarding the snow cover variability, we limited the representation to the behavior of the entire study area to daily, monthly, and yearly resolutions referred to the cold season (April–October). The snow days (season duration) had a good performance (Pearson's correlation of 0.9) compared with the number of skiing days at Cerro Mirador presented by Aguirre et al. (2018) [3], with a significant negative-snow-days trend of 0.5 days/year at Cerro Mirador in 2000–2020. For the whole of the Brunswick Peninsula, the snow days showed a significant positive trend of 0.54 snow day/year for the 2000–2020 period, while a strong significant decreasing trend of $-4.64$ snow day/year was detected for the last 10 years (2010–2020). In a long-term analysis, the work of Aguirre et al. (2018) showed a significant decreasing trend of 19% of snow cover in the Brunswick Peninsula for the last 45 years (1972–2016), which could be attributed to a statistically significant long-term warming of 0.71 °C in Punta Arenas during winter.

*5.2. Climatic Forcing*

The spatio-temporal model shown in Figure 13, which represents both the yearly snow-days trend and snow days as an annual average (2000–2020), explains 22.7% and 64.7% of the data deviance, respectively. The most affected areas regarding snow cover reduction are those located within the elevation band below 400 m a.s.l. In an earlier study [3], an unequivocal forcing of the snow cover reduction occurred due to the regional temperature increase in southern Patagonia. A significant warming temperature of 0.72 °C (0.12 °C/decade) was detected in 1958–2016 during the winter season [3], based on the data of the Punta Arenas Airport weather station. The work by Srur et al. (2018) [92] in Patagonia (49.5°S/73°W) points to a similar observation, where a displacement of the tree line (Nothofagus pumilio) to higher elevations is detected, also pointing to atmospheric warming.

To study the climate in remote environments with a poor availability of weather stations [93–96], atmospheric model output data can be used, such as the ERA5 reanalysis dataset. This ECMWF reanalysis product covers the period from 1979 to the present, at a coarse resolution of 30 km. A higher resolution (ERA5 Land [97]) of 9 km, which has recently been used in southern Patagonia (i.e., [33,98,99]), is available. Here, we analyzed the ERA5 Land hourly temperature and total precipitation data to extend and compare climate variables with the MODIS snow reconstruction in the Brunswick Peninsula. Based on the ERA5 data, we assessed mean, minimum, and maximum monthly temperature and monthly degree hours (defined as the number of monthly hours above 0 °C). For solid and liquid precipitations, we used a threshold relation with air temperature to estimate

the solid/liquid fraction in total precipitation as indicated in the work of Schaefer et al. (2013) [100], with an upper-temperature threshold of 1.6 °C and a lower threshold of 0.9 °C, as proposed by Aguirre et al. (2018) [3]. Above 1.6 °C, only liquid precipitation occurred, while below 0.9 °C, only solid precipitation was observed, with a linear change between these two thresholds. Table 7 shows the Pearson's cross-correlations between monthly snow cover area (km²), reconstructed from the MODIS data, and the ERA5 Land climate variables for the cold season (April to October).

**Table 7.** Pearson's cross-correlation between monthly snow cover area from MODIS data at Brunswick Peninsula and ERA5 land monthly variables from ERA5 data at AWS Tres Morros station for the period of 2000 to 2020, between April to October. ERA 5 variables are liquid and solid precipitation levels; mean, minimum, and maximum temperatures; and monthly degree hours.

| ERA5 Climate Variables | Liquid Precipitation | Solid Precipitation | Mean Temperature | Maximum Temperature | Minimum Temperature | Degree Hours |
|---|---|---|---|---|---|---|
| Cross Pearson correlation with MODIS snow cover area | −0.49 | 0.48 | −0.70 | −0.65 | −0.60 | −0.70 |

This result confirms that the main climatic driver for snow cover reduction is related to a mean atmospheric temperature increase, or degree hours, both of which show a Pearson's cross-correlation of −0.70. Linear regressions of the monthly mean atmospheric temperature and MODIS snow cover area (Snow_cover ~ 72.43–14.61 (t_mean)) show a determination coefficient $R^2$ of 0.48 ($p < 0.001$). We then adjusted a linear and non-linear GAM model to explain the MODIS snow area based on the mean atmospheric temperature, liquid precipitation, and solid precipitation. In those two models, only the mean temperature proved to be significant, while the other two variables (liquid and solid precipitations) did not result in a significant relation to snow cover extent.

The regional climate model RegCM4 at a resolution of 10 km was run for all of Chile: hindcast, from 1979 to 2015; historical, from 1975 to 2005; and modeling near future, from 2020 to 2050. Two different climate change scenarios have been considered: RCP 2.6 and 8.5 [101]. Figure 14a shows the daily temperature correlations ($R^2 = 0.86$) between ERA5 Land product and AWS Tres Morros data in 2018–2020. This correlation increases significantly to $R^2 = 0.98$ on a monthly scale (Figure 14b). With this validation, we used ERA5 land temperature data to check the performance of RegCM4 at AWS Tres Morros (2000–2005), obtaining $R^2 = 0.53$, which the GAM model adequately represented due to its non-linear behavior (Figure 14b).

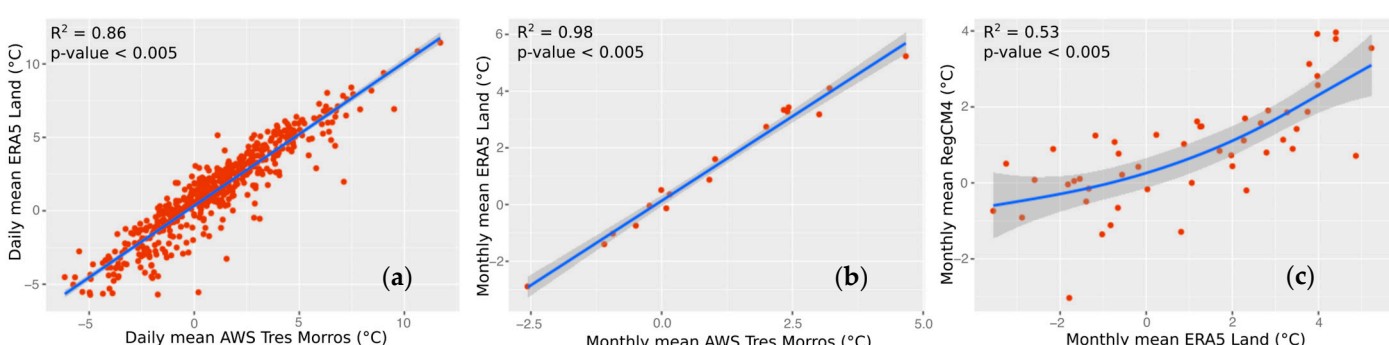

**Figure 14.** Reanalysis and regional climate model validation. (**a**) Scatter plot of daily mean temperatures between ERA5 Land and AWS Tres Morros (2018–2020); (**b**) scatter plot of monthly mean temperatures between ERA5 Land and AWS Tres Morros (2018–2020); and (**c**) scatter plot of monthly mean temperatures between ERA5 Land and RegCM4 (2000–2005).

The trend in the ERA5 Land monthly mean temperature extracted from the location at AWS Tres Morros shows significant warming in May (0.068 °C/year) and October (0.098 °C/year) for the 2000–2020 period, which coincide with the start and end of the cold season. For the future (2020–2050) scenario RCP 8.5 in the Brunswick Peninsula, we used the regional climate model RegCM4 corrected by ERA5 Land, considering the relationship observed in Figure 14c. This scenario presents a significant warming trend during July (0.059 °C/year), August (0.088 °C/year), and October (0.019 °C/year), which predicts that snow cover reduction will continue in Patagonia.

## 6. Conclusions

This work presented a novel and robust methodology for a high-resolution spatio-temporal snow cover reconstruction in southern Patagonia. Our results include a semi-automatic integration of three main processes: (1) increase in the spatial resolution of MODIS data to 250 m by means of a new relationship that includes snow cover; (2) a new proposal of snow–cloud discriminations; and (3) a daily spatio-temporal reconstruction of snow extent and its albedo at the subpixel level (250 m), validated in the Brunswick Peninsula.

The results show a significant decreasing trend in the snow season of 4.64 days/year in the period 2010–2020. The most affected elevation is below 400 m, with a non-significant negative trend about −0.5 snow days per year, mainly due to warming at the beginning and end of the snow season (May and October). The projection under the future RCP8.5 scenario, using the results from the RegCM4 regional climate model at a resolution of 10 km, shows significant warming of 0.059 °C/year during July, 0.088 °C/year during August, and 0.019 °C/year during October in the 2020–2050 period, in contrast to the study period (2000–2020) where ERA5 Land shows significant warming in May (0.068 °C/year) and October (0.098 °C/year) in the Brunswick Peninsula.

Given the results of this work, it is important to improve the quantification of snow cover (i.e., fSC, snow depth, snow water equivalent), especially in vegetated areas, such as in the low–mid elevations in Patagonia, since relevant snow underestimations can occur. Furthermore, due to the relevance of snow cover for human development, water resources, and the environment, its evolution under future climate change scenarios should be studied in detail, deploying a robust ground-based monitoring data collection, over a broader territorial representation.

**Supplementary Materials:** The following supporting information can be downloaded at: https://www.mdpi.com/article/10.3390/rs15225430/s1, Figure S1: Monthly trend of snow cover area in the Brunswick Peninsula; Figure S2: PCA of snow-days average and trend; Figure S3: Generalized additive models to Mod_av; Figure S4: Generalized additive models to Mod_trend. In the Supplementary Materials we add all the data used in this study: https://drive.google.com/file/d/1gBQApwfkKjlj8tTtfnhPCkzDKZesEFfx/view?usp=sharing, accessed on 1 December 2021.

**Author Contributions:** Conceptualization, F.A., R.J. and G.C.; methodology, F.A., R.J. and G.C.; software, F.A.; validation F.A., D.B., T.S., J.C., R.J. and G.C.; formal analysis, F.A., R.J. and G.C.; investigation, F.A., R.J. and G.C.; resources, R.J.; data curation, F.A.; writing—original draft preparation, F.A.; writing—review and editing, F.A., D.B., J.C., T.S., C.S., R.J. and G.C.; visualization, F.A.; supervision, R.J. and G.C.; project administration, R.J.; funding acquisition, R.J. and G.C. All authors have read and agreed to the published version of the manuscript.

**Funding:** This study was co-funded by the FNDR GORE Magallanes "Programa de Transferencia Científico Tecnológico Modelamiento Climático Planificación, XII Región, Código BIP N° 30462410", known as the MoCliM Programme, executed by the Chilean Antarctic Institute, with the collaboration of Universidad de Magallanes. Proyecto MIAS from the Innovation Directorate of the Universidad de Magallanes funded the publication costs.

**Data Availability Statement:** In the Supplementary Materials we add all the data used in this study. To MODIS data and for optimize the download process, we used the script "order_MWS.pl", available from: https://ladsweb.modaps.eosdis.nasa.gov/tools-and-services/, accessed on 8 August 2023. In

Supplementary Materials we mentioned the different scripts used. Also, we include the data from AWS Tres Morros for the period analyzed. As a future work we will systematize the scripts on an open platform like GitHub.

**Acknowledgments:** We appreciate the support and information provided by the Corporación Nacional Forestal (CONAF) of Magallanes and Club Andino of Punta Arenas. The collaboration with Inti González and Rodrigo Adaros during the field work is appreciated. Acknowledges support from ANID-FONDAP-1522A0001 and COPAS COASTAL ANID FB210021. Powered@NLHPC: This research was supported by the supercomputing infrastructure of the NLHPC (ECM-02). We are also grateful for the contributions made by the Cape Horn International Centre (CHIC), ANID/BASAL FB210018, and Gaia Antarctica Research Centre (CIGA) of Universidad de Magallanes. The authors gratefully acknowledge the valuable comments and suggestions from both reviewers and the editor that helped improving the manuscript.

**Conflicts of Interest:** The authors declare no conflict of interest.

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
