# Peer review of "Snow Cover Reconstruction in the Brunswick Peninsula, Patagonia, Derived from a Combination of the Spectral Fusion, Mixture Analysis, and Temporal Interpolation of MODIS Data"

_remotesensing, doi:10.3390/rs15225430_

Round 1
Reviewer 1 Report
Comments and Suggestions for Authors
Figures, tables and formats in the article need to be uniformly adjusted in accordance with the journal's standards. Detailed questions and suggestions are as follows:
Comment 1: Equation 1 in the Introduction should be placed in the method.
Comment 2: There are two obvious citation errors in the study area section, "The city of Punta Arenas, located in the north-eastern sector of the Peninsula, concentrates 80% of the population of the Magallanes province, one of the areas with the lowest population density in Chile (1.1 inhabitants/km2 ) 2 " and "Snow cover has shown a relevant loss over the last several decades, mainly due to atmospheric warming (Aguirre et al., 2018)".
Comment 3: The right figure of Figure 1 does not show the scale, and a and b need to be added to distinguish between the two figures.
Comment 4: The subheadings in Section 3.1 are not standardized and should be denoted by 3.1.1 and 3.1.2 respectively.
Comment 5: The format of Equation 10 is not standardized and should be centered.
Comment 6: In Section 3.2.4, it is shown that MODIS passes around 11:00h local time, but the selected verification data is the daily average of 11:00, 12:00, 13:00, why not choose 10:00, 11: 00, 12:00 daily average?
Comment 7: Figure 3 needs a scale bar.
Comment 8: The format of Table 5 is not standardized, and the header of each column needs to be added.
Comment 9: Figure 4 needs to add color bar.
Comment 10: Figure 5, Figure 7, and Figure 9 use UTM coordinates, while other figures in the article use WGS84 coordinates, and the full text needs a unified coordinate system.
Comment 11: In Table 7, there is no correspondence between the season and the image time.
Comment 12: On page 23, "In Figure 7 the results of the SMA are presented. Figure 7-B shows snow fraction (%) and Figure 7-D snow grain size (μm), compared with the topographic elevation (Fig. 7-A). For broadband albedo calculation refer to Script Nº 4 in methodology and Appendix A." cannot be an independent paragraph, it should be the explanation of Figure 7, which needs to be deleted here.
Comment 13: In the image on August 6 in Figure 8, it can be seen that there is a mist, and the distribution of snow can be seen under the mist. Is there no further treatment of the haze in the text? If not, further processing is required, which will increase the accuracy of the results.
Comment 14: On the 28 pages, “98%” do not need to be bolded.
Comment 15: Obvious segmentation fault occurred on page 32, "It would be interesting for future applications of this methodology create a long-time series to provide a more extensive assessment for all the images processed but for our study this is out of the main scope since it is the first time this technique is implemented over this region.
Of high importance are cloud and snow discrimination. ".
Comment 16: Add literature to "such as the ERA5 reanalysis dataset" on page 33:
“Cao B, Gruber S, Zheng D, et al. The ERA5-Land soil temperature bias in permafrost regions[J]. The Cryosphere, 2020, 14(8): 2581-2595.”
“Cao B, Arduini G, Zsoter E. Brief communication: Improving ERA5-Land soil temperature in permafrost regions using an optimized
Author Response
Thank you very much for your comments and suggestions, which certainly helped to improve the manuscript. In the attached, we proceed to answer each of the comments and suggestions. Answers and new text are shown in blue in the new version of the manuscript.

Reviewer 2 Report
Comments and Suggestions for Authors
The change of snow cover is related to climate change. This paper studies the snow cover variability in Brunswick Peninsula, Patagonia, from the satellite remote sensing data (MODIS data).
However, the methodology is not clearly explained. The expressions for the methodology from the title, the abstract and "3.2 Methods" are not consistent. From the title, it is "a combination of spectral fusion, mixture analysis and temporal interpolation"; from the abstract, "This work presents a novel methodology that integrates three indexes applied to MODIS satellite data"; from "3.2 Methods", several methods are introduced, but the combination is not clearly explained. Is it possible to use a flow chart to explain the methodology?
In the abstract, there is an expression "a novel methodology that integrates three indexes", however, how to integrate the three indexes is not clearly explained in "3.2 Methods". Is it possible to use a formula to represent the integration?
The novelty or innovation of the methods is not clearly explained or argued.
The improvement or difference from the existing methods is not clearly explained or argued.
In "3.2 Methods", three methods are introduced: "3.2.1 Downscaling and spectral fusion", "3.2.2 Spectral mixture analysis", and "3.2.3 Spatio-temporal snow reconstruction".
In "3.2.1 Downscaling and spectral fusion", Line 237 "We use different indexes for the quality assessment and selection of the downscaling methodology", but why to use different indexes (or the advantage for using different indexes) is not clearly argued.
In "3.2.2 Spectral mixture analysis" and "3.2.3 Spatio-temporal snow reconstruction", the difference or improvement from the existing methods is not clearly explained.
In "4 Results", Figure 11 shows an obvious four-year increase of snow cover area from 2017 to 2020. Is it possible to extend the end year to 2022? If the snow cover area increases from 2020 to 2022, will the conclusion for the last 10 years (2012-2022) be different from the conclusion "However, for the last 10 years (2010-2020), a significant decreasing trend of -4.64 snow days/year is observed in Brunswick Peninsula"?
Because the change of snow cover is related to climate change, is it possible to give the relationship or formula between snow cover variability and temperature change?
Specific comments
--- In the abstract, from the sentence "This work presents a novel methodology that integrates three indexes applied to MODIS satellite data (Spectral Mixture Analysis (SMA), Normalized Difference Snow Index (NDSI) and Melt Area Detection Index (MADI))", it seems "Spectral Mixture Analysis (SMA)" is an index, however, in "3.2 Methods", SMA is described as a method. The concept is not consistent.
--- Line 40-41, "since the surface temperatures are close to the melting point", why?
--- Line 59-60,"The basic algorithm (Eq. 1) for generating the operational snow cover product from MODIS data is the Normalized Difference Snow Index (NDSI)". Is the index algorithm? Is the concept consistent?
--- Line 219, the symbol ?_? in (Eq. 4) is not explained.
--- Line 240, the symbol p in (Eq. 5) is not explained.
--- Line 295-296, the statement "? uses a relationship between the illumination angle (defined as solar zenith angle – terrain slope) ?0 and the grain size of the snow detected, ? in the SMA for each pixel" is not consistent with (Eq. 11).
--- Line 329, "we use a time window of 15 days". Why to use a time window of 15 days, not other number of days? is it optimal? It is not argued.
--- Line 627, "between SF_Rec ~ s(aws_sh) " is confusing.
--- Line 643, "45 years (2000-2016) " is difficult to understand.
Comments on the Quality of English Language
--- Line 39, in "various Earth System components", "System"-->"system"?
--- The capitalization is not consistent in different sentences. For example, Line 267 "The spectral mixture analysis (SMA)", Line 272 "Multiple Endmember Spectral Mixture Analysis (MESMA)", and so on.
--- Line 278, in "(o better said a pixel’s spectral response)", "o"-->"or"?
--- Line 614-616, in "It would be interesting for future applications of this methodology create a long-time series to provide a more extensive assessment for all the images 615 processed but for our study this is out of the main scope since it is the first time this technique is implemented over this region.", the use of "create' seems a grammar problem.
If a comma is added before "but", it would be easier to read.
--- Line 629, in "Of a total of 159 days of", too many "of" are used in this phrase.
--- Line 731, a period "." is needed at the end of this sentence.
Author Response

(The authors gave the same response as above.)

Reviewer 3 Report
Comments and Suggestions for Authors
Under the background of climate change and extreme weather events, snow cover becomes an important index for climate change and water resource management. The fractional snow cover dataset is very important for cryosphere science and its related disciplines. This study proposes a novel and robust methodology for a high-resolution spatio-temporal snow cover reconstruction in southern Patagonia.
I think the content of the original manuscript is sufficient and the description is clear for understanding. However, I still have some comments which should be noticed by the authors.
(1) The introduction does not describe the current challenge and progress on the FSC retrieval from remote sensing data.
(2) Is only one ground meteorological station used to verify the accuracy of improved products? I think the validation results are not convincing, and I suggest using high-resolution remote sensing data (such as Landsat) for further validation.
(3) Please write more in detail about the climate of the study area, eg., temperature, precipitation, humidity, etc.
(4) In this study, a threshold of FSC > 20% is used to consider a pixel as snow-covered. I suggest discussing the impact of different FSC thresholds on accuracy verification and result analysis.
(5) In discussion section, what is the correlation between the analysis of different climate change scenarios and the content of this study?
(6) What is the novelty of this study compared to Wang et al. (2015)?
(7) Please clearly state the limitation of this study and future work.
(8) My other issue is the length of the text, the content in Methods should be briefly explained.
Author Response

(The authors gave the same response as above.)

Round 2
Reviewer 2 Report
Comments and Suggestions for Authors
There are some questions here.
--- In the abstract, does "a novel methodology" mean "The snow cover assessment method"?
--- What is the relationship between "Snow cover reconstruction" in the title and "The snow cover assessment method" in the abstract?
--- Abstract, in "This work evaluates and proposes a novel methodology that integrates...", what does "evaluates" mean? Does it mean the comparison with the existing methods? Is it possible to add explanation for this?
--- In "3 Data and methods", "In this section, we describe the different data and methods used to reconstruct the spatio-temporal variability of snow in southernmost Patagonia." This expression is not consistent with the phrases "a novel methodology" and "The snow cover assessment method" in the abstract?
--- In "3 Data and methods", "The novel characteristic of the methodology developed here is the automatic integration of three main methods: ...", what does “automatic” mean? Does it just put the scripts together and run together?
--- For "Figure 2: General diagram of the snow cover reconstruction method", is it possible to add sentences to explain how the three methods are integrated? Does the integration just put the three methods in series?
--- In "6 Conclusions", is it possible to add a summary to explain briefly how the three methods are integrated? is it possible to add a summary to explain briefly how "a novel methodology" is evaluated?
--- Line 753, in "MODIS Data dawload", is "dawload" a typo?
--- In "8 Appendix A. Supplementary data and codes", Scripts Nº1 - Nº5, is there a web address for these scripts?
--- It suggests that authors carefully check the whole manuscript for inconsistent phrases or expressions, as well as typos. If possible, try to improve the manuscript.
Author Response
Thank you very much for your comments and suggestions, which certainly helped to improve the manuscript. Below we proceed to answer each of the comments and suggestions in blue.
Also, these changes are in blue in the main manuscript.
My best wishes.
